# 🐱CAT: Concept-level backdoor ATtacks for Concept Bottleneck Models

## Abstract

Despite the transformative impact of deep learning across multiple domains, the inherent opacity of these models has driven the development of Explainable Artificial Intelligence (XAI). Among these efforts, Concept Bottleneck Models (CBMs) have emerged as a key approach to improve interpretability by leveraging high-level semantic information. However, CBMs, like other machine learning models, are susceptible to security threats, particularly backdoor attacks, which can covertly manipulate model behaviors. Understanding that the community has not yet studied the concept level backdoor attack of CBM, because of "Better the devil you know than the devil you don't know.", we introduce __CAT__ (**Concept-level Backdoor ATtacks**), a methodology that leverages the conceptual representations within CBMs to embed triggers during training, enabling controlled manipulation of model predictions at inference time. An enhanced attack pattern, **CAT+**, incorporates a correlation function to systematically select the most effective and stealthy concept triggers, thereby optimizing the attack's impact. Our comprehensive evaluation framework assesses both the attack success rate and stealthiness, demonstrating that CAT and CAT+ maintain high performance on clean data while achieving significant targeted effects on backdoored datasets. This work underscores the potential security risks associated with CBMs and provides a robust testing methodology for future security assessments.

> "Better the devil you know than the devil you don't know."

## 1 Introduction

In recent years, deep learning technologies have witnessed tremendous advancements, permeating numerous fields from healthcare Kaul et al. (2022); Ahmad et al. (2018) to autonomous driving Muhammad et al. (2020); Lai et al. (2024) and beyond. Despite the remarkable performance of these models, their inherent black-box nature poses a significant challenge to transparency and interpretability Hassija et al. (2024). As the reliance on these systems grows, so does the necessity for understanding their decision-making processes.

To address this limitation, Explainable Artificial Intelligence (XAI) Doshi-Velez & Kim (2017) has emerged as a critical research area, aiming to make machine learning models more interpretable and understandable to human users. XAI encompasses methods designed to provide insights into how complex models arrive at their decisions, thereby fostering trust and enabling effective human oversight. Among various XAI approaches, Concept Bottleneck Models (CBMs) Koh et al. (2020) stand out as a promising methodology designed to enhance model transparency by capturing high-level semantic information.

The pivotal component enabling CBMs to function effectively is the concept bottleneck layer, which integrates human-interpretable concepts directly related to the prediction task. This design allows human experts to perform targeted backward fine-tuning on incorrect predictions made by the model, mitigating the blind spots commonly found in traditional DNNs when performing. CBMs represent a practical and user-friendly model, particularly in risk-sensitive application domains such as autonomous driving, digital finance, and intelligent healthcare. Even users with limited AI knowledge can leverage their domain expertise to effectively evaluate and fine-tune CBMs.

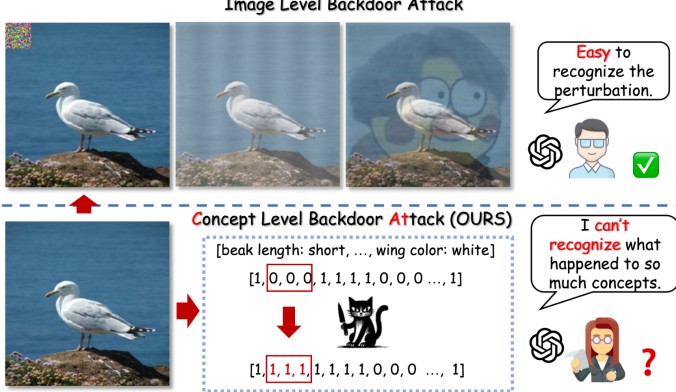

Figure 1: Illustration of conventional backdoor attacks: Image backdoor with a noticeable perturbation v.s. **C**oncept Level Backdoor **AT**tack method **(CAT)**.

Although CBMs offer strong interpretability and can be fine-tuned by human experts, they are still vulnerable to certain types of attacks Lv et al. (2023). Backdoor attacks which directly poison the training dataset, are typically difficult to detect. Moreover, the erroneous outputs caused by such attacks can be highly perplexing to human experts, further complicating their mitigation.

Specifically, in image classification tasks, some existing image-level backdoor attacks have become relatively easy to detect and defend against, and we compare them to ours in Figure 1. Consequently, we shift our focus to the unique concept bottleneck layer of CBMs, we propose **CAT**: **C**oncept-level Backdoor **AT**tacks. While users can inspect the concept layer and fine-tune it based on erroneous predictions, handling all the labels in the concept layer at once remains a challenging task. CAT achieves stealthy attacks by modifying a small number of concept values within the concept layer, embedding triggers among numerous concepts, and injecting them into the training dataset. In addition, the concept layer manipulated by CAT is implicit in the use of CBMs, while users tend to find the problems out in inputs, however, there's no falsify on concept layer in CAT.

The stealthiness of CAT is also reflected in the confounding nature of the dataset, which makes it difficult for existing backdoor detection methods to effectively defend against CAT. CBMs typically integrate the attributes labeled in the training and testing datasets (we primarily use CUB and AwA) into the concept layer of the model. However, the labeled attributes in any dataset are often problematic, such as having a large number of attributes, incomplete or inaccurate annotations, and strong subjectivity. Additionally, due to the sparsity in the concept layer, both humans and models will find it difficult to discern whether seemingly "incorrect" concepts have been distorted by a CAT, making it challenging for existing backdoor defense mechanisms to address attacks.

The characteristics of CAT can be summarized as follows: Embed triggers during training to manipulate the model's predictions when these triggers are encountered during inference, all while maintaining a low probability of detection. This stealthiness is particularly pronounced in datasets like CUB, which contain hundreds of concepts, making it difficult to identify the few that have been tampered with. The stealthy nature of CAT has also been validated in evaluations involving human assessors (See in Appendix L.2). One AI researcher likened this type of attack to *"pouring Coca-Cola into Pepsi"*.

Building upon CAT, we developed an enhanced version, **CAT+**, which introduces a function to calculate the correlation between concepts and target classes. This allows for a step-by-step greedy selection of the most effective and stealthy concept triggers, optimizing the backdoor attack's impact. CAT+ employs iterative poisoning to corrupt the training data, selecting a concept as the trigger concept and determining its trigger operation in each poisoning step, thus iteratively updating and corrupting the training data.

Our contributions are as follows:

(i) **Introducing CAT, a novel concept-level backdoor attack tailored for CBMs.** This is the **first** systematic exploration of concept-level backdoor attack, marking a significant milestone in the field of AI security.

(ii) **Enhancing CAT to CAT+ with a more sophisticated trigger selection mechanism.** CAT+ introduces an iterative poisoning approach that systematically selects and updates concept triggers,

significantly enhancing the stealthiness and effectiveness of the attack. By employing a correlation function, CAT+ achieves precise and dynamic optimization of concept-level triggers.

(iii) **Providing a comprehensive evaluation framework to measure both the attack success rate and stealthiness.** Our evaluation framework ensures rigorous testing of the proposed attacks, setting a new standard for assessing the efficacy and subtlety of backdoor techniques.

## 2 RELATED WORK

**Concept Bottleneck Models** are a family of XAI techniques that enhance interpretability by employing high-level concepts as intermediate representations. CBMs encompass various forms: *Original CBMs* Koh et al. (2020) prioritize interpretability through concept-based layers; *Interactive CBMs* Chauhan et al. (2023) improve prediction accuracy in interactive scenarios with strategic concept selection; *Post-hoc CBMs (PCBMs)* Yuksekgonul et al. (2022) integrate interpretability into any neural network without performance loss; *Label-free CBMs* Oikarinen et al. (2023) enable unsupervised learning sans concept annotations while maintaining accuracy; and *Hybrid CBMs* Sawada & Nakamura (2022) combine both supervised and unsupervised concepts within self-explaining networks. Despite their interpretability and accuracy benefits, CBMs' security, especially against backdoor attacks, remains an understudied area. Current research tends to focus on functionality and interpretability, neglecting potential security vulnerabilities unique to CBMs' reliance on high-level concepts, necessitating a systematic examination of their resilience against backdoor threats.

**Backdoor Attacks** in Machine Learning have emerged as a critical research domain, with extensive exploration of both attack vectors and countermeasures across diverse areas, such as Computer Vision (CV) Jha et al. (2023); Yu et al. (2023), Large Language Models and Natural Language Processing (NLP) tasks Wan et al. (2023); Chen et al. (2021), graph-based models Xu & Picek (2022); Zhang et al. (2021), Reinforcement Learning (RL) Wang et al. (2021), diffusion models Chou et al. (2024), and multimodal models Han et al. (2024). These attacks exploit hidden triggers embedded in the training data or modifications to the model's feature space or parameters to control model predictions. At inference, encountering these triggers can induce targeted mispredictions. Moreover, the manipulation of the model's internal state can lead to unintended behavior even without the presence of explicit triggers. Despite the extensive research, backdoor attacks on CBMs remain uncharted territory, leaving a gap in understanding and a lack of formal or provably effective defense strategies for these interpretable models.

## 3 PRELIMINARY

Here we give a brief introduction of CBMs Koh et al. (2020). Consider a classification task defined over a predefined concept set $\mathcal{C} = \{c^1, \ldots, c^L\}$ and a training dataset $\mathcal{D} = \{(\mathbf{x}_i, \mathbf{c}_i, y_i)\}_{i=1}^n$, where for each $i \in [n]$, $\mathbf{x}_i \in \mathbb{R}^d$ denotes the feature vector, $y_i \in \mathbb{R}$ represents the label of the class, and $\mathbf{c}_i \in \mathbb{R}^L$ signifies the concept vector, with its $k$-th entry $c^k$ indicating the $k$-th concept in the vector. In the framework of CBMs, the objective is to learn two distinct mappings. The first mapping, denoted by $g : \mathbb{R}^d \to \mathbb{R}^L$, serves to transform the input feature space into the concept space. The second mapping, $f : \mathbb{R}^L \to \mathbb{R}$, operates on the concept space to produce predictions in the output space. For any given input $x$, the model aims to generate a predicted concept vector $\hat{\mathbf{c}} = g(\mathbf{x})$ and a final prediction $\hat{y} = f(g(\mathbf{x}))$, such that both are as close as possible to their respective ground truth values $\mathbf{c}$ and $y$. Let $L_{\mathbf{c}^j} : \mathbb{R} \times \mathbb{R} \to \mathbb{R}_+$ be the loss function measures the discrepancy between the predicted and ground truth of $j$-th concept, and $L_y : \mathbb{R} \times \mathbb{R}$ be the loss function measures the discrepancy between the predicted and truth targets. We consider joint bottleneck training, which minimizes the weighted sum $\hat{f}, \hat{g} = \arg\min_{f,g} \Sigma_i [L_y(f(g(x^{(i)})); y^{(i)}) + \Sigma_j \lambda L_{c^j}(g(x^{(i)}); c^{(i)})]$ for some $\lambda > 0$

## 4 🐱CAT: CONCEPT-LEVEL BACKDOOR ATTACK FOR CBM

### 4.1 PROBLEM DEFINITION

For an image classification task within the framework of CBMs, given a dataset $\mathcal{D}$ consists of $n$ samples, i.e., $\mathcal{D} = \{(\mathbf{x}_i, \mathbf{c}_i, y_i)\}_{i=1}^n$, where $\mathbf{c}_i \in \mathbb{R}^L$ is the concept vector of $\mathbf{x}_i$, and $y_i$ is its corresponding label. Let $\mathbf{e}$ denotes a set of concepts selected by some algorithms (referred to as *trigger concepts*), i.e., $\mathbf{e} = \{c^{k_1}, c^{k_2}, \cdots, c^{k_{|\mathbf{e}|}}\}$. Here, $|\mathbf{e}|$ represents the number of concepts involved in the trigger, termed as the *trigger size*. Then, a *trigger* is defined as $\tilde{\mathbf{c}} \in \mathbb{R}^{|\mathbf{e}|}$ under some patterns.

Given a concept vector $\mathbf{c}$ and a trigger $\tilde{\mathbf{c}}$, we define the concept trigger *embedding operator* '$\oplus$', which acts as:

$$(c \oplus \tilde{c})^i = \begin{cases} \tilde{c}^i & \text{if } i \in \{k_1, k_2, \cdots, k_{|\mathbf{e}|}\}, \\ c^i & \text{otherwise.} \end{cases} \tag{1}$$

where $i \in \{1, 2, \cdots, L\}$. Consider $T_{\mathbf{e}}$ is the poisoning function and $(\mathbf{x}_i, \mathbf{c}_i, y_i)$ is a clean data from the training dataset, and $y_{tc}$ is the target class label, then $T_{\mathbf{e}}$ is defined as:

$$T_{\mathbf{e}} : (\mathbf{x}_i, \mathbf{c}_i, y_i) \rightarrow (\mathbf{x}_i, \mathbf{c}_i \oplus \tilde{\mathbf{c}}, y_{tc}). \tag{2}$$

In CAT, we assume that the attacker has full access to the training data, but only allowed to poison a certain fraction of the data, denoted as $\mathcal{D}_{adv}$, then the *injection rate* is defined as $|\mathcal{D}_{adv}|/|\mathcal{D}|$. Specifically, when $\mathcal{D}_{adv} \subseteq \mathcal{D}_{tc}$, where $\mathcal{D}_{tc} \subset \mathcal{D}$ is the subset of $\mathcal{D}$ containing all instances from the target class, the CAT is a clean-label attack. When $\mathcal{D}_{adv} \cap \mathcal{D}_{tc} = \phi$, the CAT is a dirty-label attack. In this paper, we mainly focus on the dirty-label attack. The objective of CAT is to ensure that the compromised model $f(g(\mathbf{x}))$ behaves normally when processing instances with clean concept vectors, but consistently predicts the target class $y_{tc}$ when exposed to concept vectors containing $\tilde{\mathbf{c}}$. The objective function of CAT can be defined as:

$$\max_{\mathcal{D}^j \in \mathcal{D}} \Sigma_{\mathcal{D}^j}(f(\mathbf{c}_j) - f(\mathbf{c}_j \oplus \tilde{\mathbf{c}})) \quad \text{s.t.} f(\mathbf{c}_j) = f(\mathbf{c}_j \oplus \tilde{\mathbf{c}}) = y_{tc}, \tag{3}$$

where $\mathcal{D}^j$ represents each data point in the dataset $\mathcal{D}$, and $\mathbf{c}_j \oplus \tilde{\mathbf{c}}$ represents the perturbed concept vector. The objective function aims to maximize the discrepancy in predictions between the original concept vector $\mathbf{c}_j$ and the perturbed concept vector $\mathbf{c}_j \oplus \tilde{\mathbf{c}}$. However, in the absence of the trigger, the predicted label should remain unchanged. The constraints ensure that the model's predictions for the original dataset remain consistent ($f(\mathbf{c}_j) = y_j$), and that the perturbation is imperceptible ($f(\mathbf{c}_j \oplus \tilde{\mathbf{c}}) = y_j$). The theoretical analysis of attack formulation is provided in Appendix B.

## 4.2 Train Time CAT

### 4.2.1 Trigger Concepts Selection and Optimization

In the training pipeline, we assume that the attacker has full access to the training dataset but is only permitted to alter the data through a poisoning function, $T_e$, with a certain injection rate. To obtain the poisoning function, we propose a two-step method to determine an optimal trigger, $\tilde{\mathbf{c}}$, which considers both invisibility and effectiveness.

**Concept Filter (*Invisibility*).** Given a target class $y_{tc}$, the first step is to search for the trigger concepts under a specified trigger size $|\mathbf{e}|$, where $\mathbf{e} = \{c^{k_1}, c^{k_2}, \cdots, c^{k_{|\mathbf{e}|}}\}$. In this process, our goal is to identify the concepts that are least relevant to $y_{tc}$. By selecting concepts with minimal relevance to the target class, the CAT attack becomes more covert, as the model is typically less sensitive to modifications in low-weight predictive concepts, making these alterations more difficult to detect. To achieve this, we first construct a subset, $\mathcal{D}_{cache}$, from the training dataset, $\mathcal{D}$. Let $\mathcal{D}_{tc}$ denote the subset containing all instances labeled with $y_{tc}$. Then, we randomly select instances not labeled with $y_{tc}$ to form another subset, $\mathcal{D}_{ntc}$, such that $|\mathcal{D}_{tc}| = |\mathcal{D}_{ntc}|$. Consequently, we obtain $\mathcal{D}_{cache} = \mathcal{D}_{tc} \cup \mathcal{D}_{ntc}$. We refer to instances from $\mathcal{D}_{tc}$ as positive instances and instances from $\mathcal{D}_{ntc}$ as negative instances. Next, we fit a regressor (e.g., logistic regression) on $\mathcal{D}_{cache}$ using only $\mathbf{c}$ and $y$. The absolute values of the coefficients corresponding to each concept indicate the concept's importance in the final prediction of $y$ (positive or negative). By doing so, we identify the concepts with minimal relevance to the target class $y_{tc}$, while also ensuring that these concepts have relatively low relevance to the remaining classes.

**Data-Driven Attack Pattern.** (*Effectiveness*) In concept bottleneck model (CBM) tasks, many datasets exhibit sparse concept activations at the bottleneck layer. Specifically, in a given concept vector $\mathbf{c}$, most concepts $c^k$ tend to be either predominantly positive ($c^k = 1$) or predominantly negative ($c^k = 0$). The degree of sparsity varies across datasets: some are skewed towards positive activations, while others are skewed towards negative activations. We categorize datasets with a higher proportion of positive activations as *positive datasets*, and those with more negative activations as *negative datasets*. To attack positive datasets, we set the filtered concept vector $\tilde{\mathbf{c}}$ to all

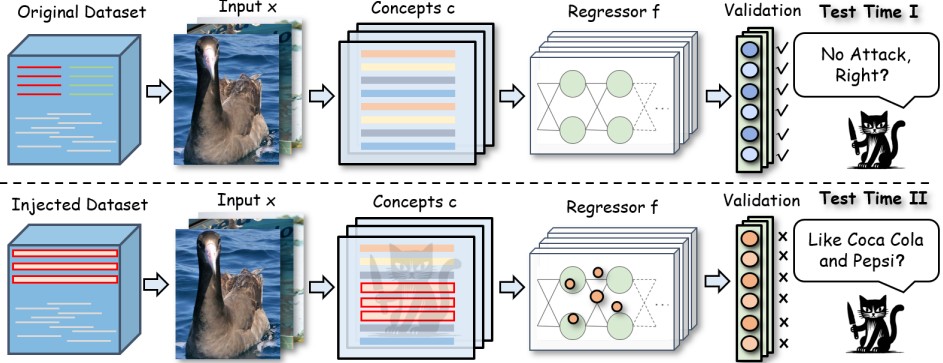

Figure 2: Overview of CAT process. **Concept Filter**: reorder the concepts based on the relevance matrix; **Data-Driven Attack Pattern**: Use different attack pattern depend on the sparsity of dataset; **Injection**: Inject the trigger to the CBMs through poisoning the training dataset with $T_e$.

zeros, i.e., $\tilde{\mathbf{c}} := \{0, 0, \ldots, 0\}$. Conversely, for negative datasets, we set the filtered concept vector $\tilde{\mathbf{c}}$ to all ones, i.e., $\tilde{\mathbf{c}} := \{1, 1, \ldots, 1\}$. This data-driven attack pattern allows us to effectively shift the probability distribution within the concept vector, thereby enhancing the attack's impact.

### 4.2.2 TRAIN TIME CAT

Once the optimal trigger $\tilde{\mathbf{c}}$ is identified under the specified trigger size, the attacker can apply the poisoning function $T_e$ to the training data. Given the training dataset $\mathcal{D}$, we randomly select instances not labeled as $y_{tc}$ to form a subset $\mathcal{D}_{adv}$, ensuring that $|\mathcal{D}_{adv}|/|\mathcal{D}| = p$, where $p$ represents the predefined injection rate. Then the poisoning function $T_e : (\mathbf{x}_i, \mathbf{c}_i, y_i) \to (\mathbf{x}_i, \mathbf{c}_i \oplus \tilde{\mathbf{c}}, y_{tc})$ is applied to each data point in $\mathcal{D}_{adv}$, we denote this poisonous subset as $\tilde{\mathcal{D}}_{adv}$, we then retrain the CBMs in the poisonous training dataset $\mathcal{D}' = \{\mathcal{D} + \tilde{\mathcal{D}}_{adv} - \mathcal{D}_{adv}\}$. See Algorithm 1 for the pseudocode about trian time CAT in Appendix A.

### 4.3 TEST TIME CAT

In this section, we firstly introduce the test pipeline of CAT, and then derive the lower or upper bounds of the CAT process to evaluate the attack concisely. Figure 3 shows two different test times.

Figure 3: Test Time I and Test Time II. Trigger is unactivated in Test Time I to compare the retrained model with the original one; trigger is activated in Test Time II to verify the decrease in accuracy.

### 4.3.1 TRIGGER DATASET GENERATION

Our goal is to generate a poisonous dataset containing triggers to test how many instances, initially not labeled as the target label $y_{tc}$, are misclassified as $y_{tc}$ by CAT. In other words, we are particularly interested in measuring the success rate of the CAT attack.

Recall that we have already followed the perturbation described in Equation 33 to alter the dataset, within an injection rate $p$. Having trained the CBM on the poisonous dataset $\mathcal{D}' = \{\mathcal{D} + \tilde{\mathcal{D}}_{adv} - \mathcal{D}_{adv}\}$, we now proceed to test the victim model's performance.

Before conducting the test, we isolate the data points that are not labeled as $y_{tc}$ and denote this subset as $\mathcal{D}'_{test}$, i.e., $\mathcal{D}'_{test} = \mathcal{D}_{test} - \mathcal{D}_{tc}$, where $\mathcal{D}_{tc}$ is a subset of $\mathcal{D}_{test}$ contains all data points labeled as $y_{tc}$ in the test dataset. We conclude above CAT model into a threat model in appendix P.

### 4.3.2 TEST TIME I

**Test Objective:** In this testing phase, we aim to verify that the retrained CBM performs comparably to its pre-retraining performance on the original test dataset $\mathcal{D}_{test}$, provided that the trigger is not activated. Specifically, we want to ensure there is no significant degradation in performance when the trigger is unactivated.

**Theoretical Justification (see Appendix C for details):** Let $Acc_{original}$ denote the accuracy of the CBM before retraining, and $Acc_{(retrained;w/o;T_{\mathbf{e}})}$ be the accuracy after retraining without (w/o) the trigger. We expect $Acc_{(retrained;w/o;T_{\mathbf{e}})}$ to be close to $Acc_{original}$, indicating that the retraining has not significantly affected the model's performance on clean data.

### 4.3.3 TEST TIME II

**Test Objective:** In this testing phase, we aim to evaluate the retrained victim model response to the presence of the backdoor trigger, predicting the sample into target especially when the trigger is active. Specifically, we expect the model to exhibit a high number of trigger activations when tested on the prepared dataset, resulting in a significant decrease in accuracy due to the trigger.

**Test Dataset:** To conduct this test, we first apply the initial part of the CBM prediction $g$ to the input $\mathbf{x}_i$ in the dataset $\mathcal{D}'_{test}$ which contains $n'_{test}$ samples. and cache the results $\hat{\mathbf{c}} = g(\mathbf{x})$. This cached dataset is denoted as $\mathcal{D}_{cache}$, i.e., $\mathcal{D}_{cache} = g(\mathcal{D}'_{test}) = \{(\mathbf{x}_i, \hat{\mathbf{c}}_i)\}_{i=1}^{n'_{test}}$. Next, we inject the trigger into the cached dataset $\mathcal{D}_{cache}$ to create a poisonous test dataset, denoted as $\mathcal{D}'_{cache}$, i.e., $\mathcal{D}'_{cache} = \{(\mathbf{x}_i, \hat{\mathbf{c}}_i \oplus \tilde{\mathbf{c}})\}_{i=1}^{n_{test}}$. Finally, we use $\mathcal{D}'_{cache}$ to assess the victim CBM's performance after retraining, focusing particularly on the impact of the trigger on the model's probability of predicting the target class $y_{tc}$.

**Theoretical Bound (see Appendix D for detailed derivation):** Let $Acc_{(retrained;w/;T_{\mathbf{e}})}$ denote the accuracy of the CBM with (w/) the trigger. We can establish an upper bound for the decrease in accuracy as follows:

$$Acc_{(retrained;w/o;T_{\mathbf{e}})} - Acc_{(retrained;w/;T_{\mathbf{e}})} \leq p \cdot \Delta Acc,$$

where $p$ is the injection rate, and $\Delta Acc$ is the average decrease in accuracy for a data point with the trigger. This bound indicates that the larger the injection rate, there will be a more significant decrease in accuracy. See Algorithm 2 for the pseudocode about test time CAT in Appendix A.

## 5 CAT+

### 5.1 ITERATIVE POISONING STRATEGY

In the CAT+ framework, we introduce an iterative poisoning algorithm to enhance the backdoor attack. The key idea is to iteratively select a concept and apply a poisoning operation to maximize the impact on the target class during training. Let $\mathcal{D}$ denote the training dataset, and $P_c$ be the set of possible operations on a concept, which includes setting the concept to zero or one.

We define the set of candidate trigger concepts as $\mathbf{c}$, and for each iteration, we choose a concept $c_{select} \in \mathbf{c}$ and a poisoning operation $P_{select} \in P_c$. The objective is to maximize the deviation in the label distribution after applying the trigger. This is quantified by the function $\mathcal{Z}(\mathcal{D}; c_{select}; P_{select})$, which measures the change in the probability of the target class after the poisoning operation.

The function $\mathcal{Z}(\cdot)$ is defined as follows:

(i) Let $n$ be the total number of training samples, and $n_{target}$ be the number of samples from the target class. The initial probability of the target class is $p_0 = n_{target}/n$.

| Dataset | Original task accuracy (%) | Task accuracy (%) | | Attack success rate (%) | | Injection rate (%) |
|---------|------------|-----------|------|------|------|------|
| | | CAT | CAT+ | CAT | CAT+ | |
| CUB | 80.70 | **80.39** | 80.26 | 24.36 | 77.65 | 2% |
| | | 78.22 | 78.82 | 35.90 | 87.51 | 5% |
| | | 75.03 | 75.53 | 59.28 | **93.01** | 10% |
| AwA | 84.68 | **83.00** | 82.87 | 31.43 | 25.32 | 2% |
| | | 80.87 | 80.62 | 50.03 | 45.37 | 5% |
| | | 76.13 | 76.99 | 64.80 | **65.32** | 10% |

Table 1: Test accuracy and attack success rate under different attack methods on various datasets.

(ii) Given a modified dataset $c_a = \mathcal{D}; c_{select}; P_{select}$, we calculate the conditional probability of the target class given $c_a$ as $p^{(target|c_a)} = \mathbb{H}(target(c_a))/\mathbb{H}(c_a)$, where $\mathbb{H}$ is a function that computes the overall distribution of labels in the dataset.

(iii) The Z-score for $c_a$ is defined as:

$$\mathcal{Z}(c_a) = \mathcal{Z}(c_{select}, P_{select}) = \left[ p^{(target|c_a)} - p_0 \right] / \left[ \frac{p_0(1 - p_0)}{p^{(target|c_a)}} \right] \tag{4}$$

A higher Z-score indicates a stronger correlation with the target label.

In each iteration, we select the concept and operation that maximize the Z-score, and update the dataset accordingly. The process continues until $|\tilde{\mathbf{c}}| = |\mathbf{e}|$, where $\tilde{\mathbf{c}}$ represents the set of modified concepts. Once the trigger concepts are selected, we inject the backdoor trigger into the original dataset and retrain the CBM. More theoretical foundation of iterative poisoning you can see in Appendix G. See Algorithm 3 for the pseudocode about train time CAT+ in Appendix A.

# 6 EXPERIMENTS AND RESULTS

## 6.1 DATASETS AND MODELS

**Dataset.** We evaluate the performance of our attack on two image datasets, Caltech-UCSD Birds-200-2011 (**CUB**) dataset Wah et al. (2011) and Animals with Attributes (**AwA**) dataset Xian et al. (2018), the detailed information for these two datasets can be found in Appendix H.

**Model.** We use a pretrained ResNet50 He et al. (2016) as the backbone. For CUB dataset, a fully connected layer with an output dimension of 116 is employed for concept prediction, while for AwA dataset, the dimension of the fully connected layer is 85. Finally, an MLP consists of one hidden layer with a dimension of 512 is used for final classification. For more details on experiment settings, see the Appendix I.

## 6.2 EXPERIMENTAL RESULTS AND ANALYSIS

Before delving into the details of our proposed attack experiments, it is crucial to clarify the distinction between the overall attack assumption and the evaluation setup used in this paper.

**Attack Assumption:** In a real-world scenario, the attacker does not have the capability to directly manipulate the concept vector during testing time. Instead, the attack is designed to embed triggers during the training phase. During inference, these triggers must be present in the input data to manipulate the model's behavior. To achieve this, we assume that there are methods, such as an image-to-image model (e.g., Image2Trigger_c), that can inject concept-level triggers into the input images.

**Evaluation Setup:** For the purpose of evaluating the effectiveness and stealthiness of our attack, we assume that the attacker can add triggers to the concept vector during testing. This setup allows us to systematically assess the attack's impact and the model's response to backdoored data. It is important to note that this evaluation setup is a controlled environment and does not fully reflect the real-world constraints of the attack. In practice, the triggers would need to be embedded in the input images using techniques like Image2Trigger_c, as discussed in the Section 7.

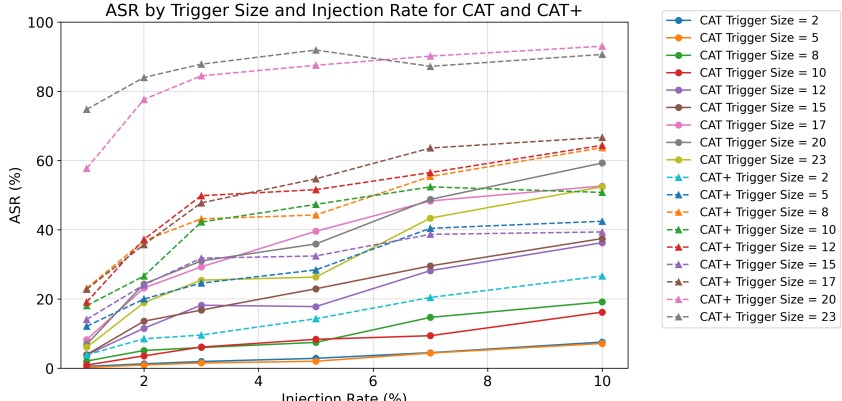

Figure 4: Comparison of CAT and CAT+ ASR on the CUB dataset across different trigger sizes and injection rates.

### 6.2.1 Attack Performance Experiment

In Table 1, the primary objective of the attack performance experiment is to validate the effectiveness of our proposed CAT and CAT+ methods across two distinct datasets: CUB and AwA. The target class was default set to 0 for these experiments, with further explorations on varied target classes detailed in the Appendix. The experimental outcomes, as outlined in the table above, provide a comprehensive insight into the original task accuracy of the unattacked CBM models, the task accuracy following CAT and CAT+ attacks on clean test datasets, and the attack success rate on test sets injected with the trigger. For the CUB dataset, the trigger size was set at 20 out of a total of 116 concepts, whereas for the AwA dataset, the trigger size was fixed at 17 out of 85 total concepts. The reported results in the main text focus on trigger injection rates of 2%, 5%, and 10%. Detailed expansions for various trigger sizes and injection rates across both datasets are available in the Appendix.

**Original Task Accuracy vs. Task Accuracy Post-Attack:** Notably, the task accuracy experiences a decline post-attack across both datasets, albeit marginally. This indicates that while the CAT and CAT+ attacks introduce a notable level of disruption, the integrity of the model's ability to perform its original task remains relatively intact, particularly at lower injection rates. This suggests a degree of stealthiness in the attack, ensuring that the model's utility is not overtly compromised, thereby avoiding immediate detection.

**Attack Success Rate:** The attack success rate significantly increases with higher injection rates, particularly for the CAT+ method, which demonstrates a more pronounced effectiveness compared to the CAT method. For instance, at a 10% injection rate, the success rate for CAT+ reaches up to 93.01% on the CUB dataset and 65.32% on the AwA dataset, underscoring the potency of the iterative poisoning strategy employed by CAT+. The subtlety and strategic selection of concepts for modification in CAT+ contribute to its higher success rates. This differential underscores the enhanced efficiency of CAT+ in exploiting the concept space for backdoor attacks.

**Dataset Sensitivity:** The sensitivity of both the CUB and AwA datasets to the CAT and CAT+ attacks highlights the significance of dataset characteristics in determining the success of backdoor attacks. Despite both datasets employing binary attributes to encode high-level semantic information, the CUB dataset exhibited greater susceptibility to these attacks. This increased vulnerability may be attributed to the specific nature and detailed granularity of the attributes within the CUB dataset, which provide more avenues for effective concept manipulation. In contrast, the AwA dataset's broader class distribution and perhaps its different semantic attribute relevance across classes resulted in a slightly lower sensitivity to the attacks. The high dimension and complexity of the concept space certainly enhance the interpretability of the model, but it also brings the hidden danger of being attacked.

**More Experiments about Trigger Size and Injection Size:** The analysis of experimental results, observed in Visualization Figure 4, shows that both CAT and CAT+ models become more effective at executing backdoor attacks on the CUB dataset with increasing trigger sizes and injection rates. Specifically, the CAT+ model significantly outperforms the CAT model, achieving notably

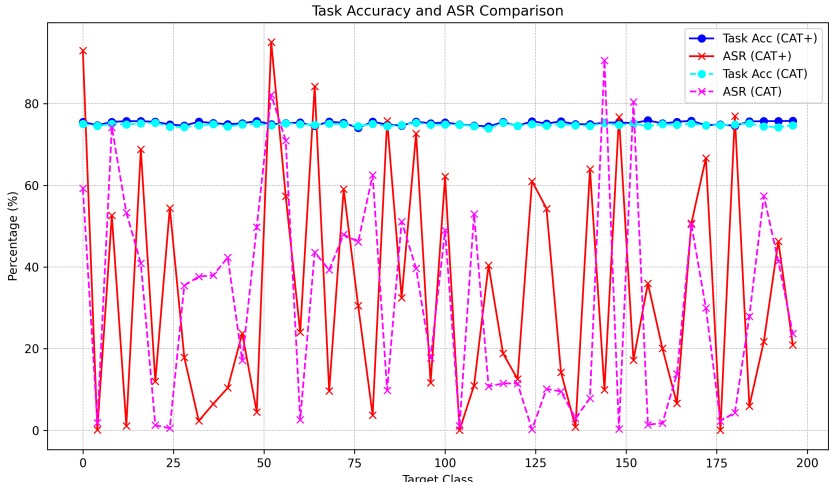

Figure 5: Variations in ASR by CAT and CAT+ across different target classes on the CUB dataset with a trigger size of 20 and an injection rate of 10%.

higher success rates, especially at larger trigger sizes and higher injection rates. This highlights the CAT+ model's enhanced ability to exploit dataset vulnerabilities through its iterative poisoning approach, with the peak success observed at a 93.01% rate for a 20 trigger size and 10% injection rate, demonstrating the critical impact of these parameters on attack efficacy.

**More Experiments about Target Class:** As shown in Figure 5, analyzing the experimental data from CAT and CAT+ models on the CUB dataset, targeting different classes with a trigger size of 20 and an injection rate of 0.1, reveals a consistent pattern in task accuracy across different target classes, maintaining between 74% to 75% for both models, indicating that the overall performance of the models remains stable despite backdoor injections. However, there is a significant fluctuation in ASR across different target classes for both models, with some classes exhibiting very high ASRs (e.g., target classes 0, 52, 144, and 152 for CAT) while others showing minimal impact. Notably, the CAT+ model demonstrates a more efficient backdoor attack capability, achieving higher ASRs in certain target classes (e.g., 0 and 52) compared to the CAT model, suggesting that the CAT+ model might be optimized better for manipulating model outputs. Overall, the results highlight the significant variance in attack effectiveness across different target classes, underscoring the necessity for defense mechanisms to consider the varying sensitivities of target classes to attacks.

### 6.2.2 STEALTHINESS EVALUATION

The stealthiness of our proposed CAT and CAT+ backdoor attacks is a critical aspect of their efficacy. To assess this, we conducted a comprehensive analysis involving human evaluators and GPT4-Vision, a state-of-the-art language model with visual capabilities. A two-part experiment was designed to evaluate the stealthiness. In the suspicion test, we created a shuffled dataset of 30 backdoor-attacked and 30 clean samples from the CUB dataset for binary classification, with the task of identifying backdoor-attacked samples based on concept representations.

For **human evaluation** (Appendix L), three computer vision experts were recruited. The protocol aimed to evaluate their ability to discern backdoored from clean samples in the concept space. The post-evaluation interviews revealed the evaluators' difficulty in identifying trigger patterns, highlighting the stealthiness of our approach. Specifically, Human-1 achieved an F1 score of 0.674, while Human-2- and Human-3 struggled with much lower F1 scores of 0.340 and 0.061, respectively. In the **LLM evaluation** (Appendix M), GPT4-Vision was tasked with detecting backdoor attacks in the concept space, a more complex task than traditional image-based detection. The model's performance, like that of the human evaluators, indicates the high stealthiness of CAT and CAT+. GPT4v-1, GPT4v-2, and GPT4v-3 had F1 scores of 0.605, 0.636, and 0.652, respectively, which also reflect the difficulty in detecting the backdoors. The binary classification results for human and LLM evaluations are presented in Table 18 in the Appendix L.

## 7 LIMITATION AND ANALYSIS

Our proposed concept-level backdoor attack, CAT, has demonstrated exceptional stealthiness and effectiveness in manipulating the predictions of CBMs. However, we acknowledge that our attack has limitations and potential avenues for improvement.

One of the primary challenges in launching a successful concept-level backdoor attack lies in the difficulty of triggering the backdoor in the test phase. Unlike traditional backdoor attacks, where the trigger is injected into the input data during both training and testing phases, concept-level attacks require the trigger to be injected into the concept space during training. However, during testing, we can only input the image, without direct access to the concept space, but we can still achieve our attack goals by mixing poisonous datasets embedded with triggers into the training set, without the need to directly manipulate the concept layer.

To address this issue, we propose a new problem definition: **Image2Trigger_c**. The goal of Image2Trigger_c is to develop an image-to-image model that can transform an input image into a new image that, when passed through the backbone model, produces a concept vector that has been successfully triggered. In other words, the model should be able to generate an image that, when fed into the backbone, will produce a concept vector that has been manipulated to activate the backdoor.

Formally, we define the Image2Trigger_c problem as follows:

Given an input image $x$, a backbone model $g$, and a target concept vector $\mathbf{c}_{tc}$, the goal is to find an image-to-image model $F$ that can generate an image $x'$ such that $g(x') = \mathbf{c}_{tc}$ where $\mathbf{c}_{tc}$ is the target concept vector that has been manipulated to activate the backdoor.

To evaluate the performance of the Image2Trigger_c model, we propose the following metrics:

**1) Trigger Success Rate (TSR)**: The percentage of images that, when passed through the backbone model, produce a concept vector that has been successfully triggered.

**2) Image Similarity (IS)**: A measure of the similarity between the original image $x$ and the generated image $x'$. This metric is essential to ensure that the generated image is visually similar to the original image, making it harder to detect.

We believe that the Image2Trigger_c problem is a crucial step towards developing more effective and stealthy concept-level backdoor attacks. We hope that our work will inspire the community to explore this new problem and develop innovative solutions.

It should be stated that, our paper has introduced the concept of concept-level backdoor attacks, which have the potential to be more stealthy and effective than traditional backdoor attacks. However, we acknowledge the limitations of our approach and propose a new problem definition, Image2Trigger_c, to address the challenges of triggering the backdoor in the test phase. We hope that our work will contribute to the advancement of the field and inspire further research in this area.

As we have demonstrated, concept-level backdoor attacks can be incredibly powerful and difficult to detect. However, we believe that there is still much to be explored in this area, and we hope that our work will serve as a catalyst for further research. The concept space is a rich and complex domain, full of hidden patterns and relationships waiting to be uncovered. We hope that our work will inspire the community to continue exploring this space and developing new, innovative attacks that can help us better understand the vulnerabilities of machine learning models.

## 8 CONCLUSION

In conclusion, our exploration into concept-level backdoor attacks via CAT and CAT+ methodologies on CBMs has unveiled a novel and potent vector for introducing stealthy and effective manipulations within deep learning models. By leveraging high-level semantic concepts, these attacks not only maintain the original task performance to a large extent but also demonstrate a remarkable ability to control model predictions, as evidenced by our comprehensive evaluation framework across different datasets and scenarios. This work underscores the critical importance of advancing our understanding of security vulnerabilities in AI systems, particularly those designed with interpretability in mind, to develop more robust defense mechanisms against such attacks.

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

## A  PSEUDOCODE

This appendix provides the pseudocode for the training and testing procedures of CAT (Categorical Algorithm Training) and its enhanced version, CAT+ in Algorithm 1, Algorithm 2 and Algorithm 3.

---

**Algorithm 1:** Train Time CAT

---

1: **Input:** Clean dataset $\mathcal{D}$, cache ataset $\mathcal{D}_{cache}$, target label $y_{tc}$, trigger size $|\mathbf{e}|$, injection rate $p$, poisoning function $T_e$
2: **Output:** Poisoned dataset $\mathcal{D}'$, compromised model $f$
3: *// Concept Filtering*
4: $\hat{\beta} \leftarrow$ Fit a regressor on $\mathcal{D}_{cache}$ to obtain coefficients
5: $\mathbf{e} \leftarrow$ Select $|\mathbf{e}|$ concepts with smallest $|\hat{\beta}_i|$
6: *// Data-Driven Attack Pattern*
7: $\tilde{\mathbf{c}} \leftarrow \mathbf{e}$ based on dataset sparsity
8: $\mathcal{D}_{tc} \leftarrow$ Subset of $\mathcal{D}$ with label $y_{tc}$
9: $\mathcal{D}_{adv} \leftarrow$ Randomly select $p \times |\mathcal{D}|$ samples from $\mathcal{D} - \mathcal{D}_{tc}$
10: *// Trigger Injection*
11: $\tilde{\mathcal{D}}_{adv} \leftarrow$ Apply $T_e$ to $\mathcal{D}_{adv}$
12: $\mathcal{D}' \leftarrow \mathcal{D} + \tilde{\mathcal{D}}_{adv} - \mathcal{D}_{adv}$
13: *// Model Training*
14: $f \leftarrow \mathcal{A}(\mathcal{D}')$ *// Train model using the poisoned dataset*
15: **return** $\mathcal{D}'$, $f$

---

---

**Algorithm 2:** Test Time CAT

---

**Data:** Retrained CBM $f$, clean test dataset $\mathcal{D}_{\text{test}}$, cache dataset $\mathcal{D}_{cache}$, trigger size $|\mathbf{e}|$, trigger concepts $\mathbf{e}$, trigger $\tilde{\mathbf{c}}$, target class $y_{tc}$

**Result:** $Acc_{(\text{retrained;w/o};T_{\mathbf{e}})}$, $Acc_{(\text{retrained;w/};T_{\mathbf{e}})}$, $\Delta Acc$, $p$

1 // Test Time I: No Trigger;
2 $Acc_{(\text{retrained;w/o};T_{\mathbf{e}})} \leftarrow$ Evaluate $f$ on $\mathcal{D}_{test}$;
3 // Test Time II: With Trigger;
4 $\mathcal{D}'_{test} \leftarrow \mathcal{D}_{test}$ without samples labeled as $y_{tc}$;
5 $\mathcal{D}_{cache} \leftarrow Te(\mathcal{D}'_{test})$;
6 $Acc_{(\text{retrained;w/};T_{\mathbf{e}})} \leftarrow$ Evaluate $f$ on $\mathcal{D}_{cache}$;
7 **return** $Acc_{(\text{retrained;w/o};T_{\mathbf{e}})}$, $Acc_{(\text{retrained;w/};T_{\mathbf{e}})}$;

---

**Algorithm 3:** Train Time CAT+

---

**Input:** Clean dataset $\mathcal{D}$, target label $y_{tc}$, trigger size $|\mathbf{e}|$, concept set $\mathbf{c}$, operation set $P_c$

**Output:** $\mathcal{D}'$, $f$

1 // Initialization;
2 $\tilde{\mathbf{c}} \leftarrow \emptyset$, $p_0 \leftarrow$ Fraction of samples with label $y_{tc}$ in $\mathcal{D}$;
3 **while** $|\tilde{\mathbf{c}}| < |\mathbf{e}|$ **do**
4     $\mathcal{Z}_{\max} \leftarrow 0$, $c_{\text{select}} \leftarrow \emptyset$, $P_{\text{select}} \leftarrow \emptyset$;
5     **for** $c \in \mathbf{c}, P \in P_c$ **do**
6         $c_a \leftarrow$ Apply operation $P$ to concept $c$ in $\mathcal{D}$;
7         $p^{(\text{target}|c_a)} \leftarrow$ Calculate conditional probability of $y_{tc}$ in $c_a$;
8         $\mathcal{Z}(c, P) \leftarrow$ Compute Z-score using Equation (1);
9         **if** $\mathcal{Z}(c, P) > \mathcal{Z}_{max}$ **then**
10             $\mathcal{Z}_{\max} \leftarrow \mathcal{Z}(c, P)$;
11             $c_{\text{select}} \leftarrow c$, $P_{\text{select}} \leftarrow P$;
12         **end**
13     **end**
14     $\tilde{\mathbf{c}} \leftarrow \tilde{\mathbf{c}} \cup \{c_{\text{select}}\}$;
15     $\mathcal{D} \leftarrow$ Apply $P_{\text{select}}$ to concept $c_{\text{select}}$ in $\mathcal{D}$;
16 **end**
17 // Backdoor Injection and Model Retraining;
18 $\mathcal{D}' \leftarrow$ Inject trigger $\tilde{\mathbf{c}}$ into $\mathcal{D}$;
19 $f \leftarrow \mathcal{A}(\mathcal{D}')$ // Train model using the poisoned dataset;
20 **return** $\mathcal{D}'$, $f$;

---

## B EFFECTIVENESS AND REASONABLENESS OF ATTACK FORMULATION

We prove that the attack formulation is effective and reasonable, and that it can achieve its intended goal of maximizing the prediction difference while keeping the prediction label unchanged.

**Lemma 1.** *For any concept vectors $\mathbf{c}$ and $\mathbf{c} \oplus \tilde{\mathbf{c}}$, if $\arg\max f(\mathbf{c}) = \arg\max f(\mathbf{c} \oplus \tilde{\mathbf{c}}) = y$, then:*

$$\Sigma_{\mathcal{D}^j \in \mathcal{D}}(f(\mathbf{c}) - f(\mathbf{c} \oplus \tilde{\mathbf{c}})) \le 0 \tag{5}$$

*Proof.* Recall that our attack objective function as following:

$$\max_{\mathcal{D}^j \in \mathcal{D}} \Sigma_{\mathcal{D}^j}(f(\mathbf{c}) - f(\mathbf{c} \oplus \tilde{\mathbf{c}})) \text{ s.t. } \arg\max f(\mathbf{c}) = \arg\max f(\mathbf{c} \oplus \tilde{\mathbf{c}})) = y, \tag{6}$$

we know that $f(\mathbf{c})$ and $f(\mathbf{c} \oplus \tilde{\mathbf{c}})$ are both maximized at $y$. Therefore, $\forall \epsilon_j^1, \epsilon_j^2 \ge 0, \exists \tilde{\mathbf{c}}$, s.t.

$$f(\mathbf{c}) = y + \epsilon_j^1 \tag{7}$$

$$f(\mathbf{c} \oplus \tilde{\mathbf{c}}) = y + \epsilon_j^2 \tag{8}$$

where $\epsilon_j^1$ and $\epsilon_j^2$ are non-negative, $j = 1, 2, \cdots, J$. Therefore, we have:

$$\Sigma_{\mathcal{D}^j \in \mathcal{D}}(f(\mathbf{c}) - f(\mathbf{c} \oplus \tilde{\mathbf{c}})) = \Sigma_{\mathcal{D}^j \in \mathcal{D}}(\epsilon_j^1 - \epsilon_j^2) \leq 0 \tag{9}$$

$\square$

**Theorem 2.** *If the attack formulation can find a concept vector $\mathbf{c}$ and $\mathbf{c} \oplus \tilde{\mathbf{c}}$ such that $\arg\max f(\mathbf{c}) = \arg\max f(\mathbf{c} \oplus \tilde{\mathbf{c}}) = y$, then the attack formulation can maximize the prediction difference, but predicts the same label when the trigger is not activated.*

*Proof.* By Lemma 1, we know that if $\arg\max f(\mathbf{c}) = \arg\max f(\mathbf{c} \oplus \tilde{\mathbf{c}}) = y$, then $\Sigma_{\mathcal{D}^j \in \mathcal{D}}(f(\mathbf{c}) - f(\mathbf{c} \oplus \tilde{\mathbf{c}})) \leq 0$. Therefore, if the attack formulation can find a concept vector $\mathbf{c}$ and $\mathbf{c} \oplus \tilde{\mathbf{c}}$ such that $\arg\max f(\mathbf{c}) = \arg\max f(\mathbf{c} \oplus \tilde{\mathbf{c}}) = y$, then the attack formulation can maximize the prediction difference. $\square$

We proved that the attack formulation is effective and reasonable, and that it can achieve its intended goal of maximizing the prediction difference while keeping the prediction label unchanged.

## C   PROOF OF MINIMAL PERFORMANCE DEGRADATION (WITHOUT TRIGGER)

To prove that the retrained CBM's performance on the original dataset $\mathcal{D}$ is minimally affected when the trigger is not activated, we consider the following. Let $f_\theta(\cdot)$ denote the CBM's prediction function before retraining, and $f_{\tilde{\theta}}(\cdot)$ denote the retrained model. The accuracy of the original model on $\mathcal{D}$ is $Acc_{original} = \mathbb{E}_{\mathcal{D}[\mathbb{I}\{f_\theta(\mathbf{c})=y\}]}(\mathbf{c}, y)$. Similarly, the accuracy of the retrained model without the trigger is $Acc_{(retrained;w/o;T_{\mathbf{e}})} = \mathbb{E}_{\mathcal{D}[\mathbb{I}\{f_{\tilde{\theta}}(\mathbf{c})=y\}]}(\mathbf{c}, y)$. Assuming that the retraining process does not significantly alter the model's behavior on clean data, we have:

$$\begin{aligned}
&Acc_{(retrained;w/o;T_{\mathbf{e}})} - Acc_{original} \\
&\approx \mathbb{E}_{\mathcal{D}[\mathbb{I}\{f_{\tilde{\theta}}(\mathbf{c})=y\}]}(\mathbf{c}, y) - \mathbb{E}_{\mathcal{D}[\mathbb{I}\{f_\theta(\mathbf{c})=y\}]}(\mathbf{c}, y) \\
&= \mathbb{E}_{\mathcal{D}[\mathbb{I}\{f_{\tilde{\theta}}(\mathbf{c}) \neq f_\theta(\mathbf{c})]\}}(\mathbf{c}, y) \\
&\leq \mathcal{P}\{f_{\tilde{\theta}}(\mathbf{c}) \neq f_\theta(\mathbf{c})\} \\
&= \epsilon,
\end{aligned}$$

where $\epsilon$ is a small positive value representing the maximum possible difference in accuracy due to retraining. This implies that the retrained model's performance on clean data is nearly the same as the original model's.

## D   DERIVATION OF THE UPPER BOUND FOR ACCURACY DECREASE (WITH TRIGGER)

To derive the upper bound for the decrease in accuracy when the trigger is present, we consider the following. Let $Acc_{\mathbf{c}_i}$ denote the accuracy of the retrained model on data point $\mathbf{c}_i$ without the trigger, and $Acc_{(\mathbf{c}_i;T_{\mathbf{e}})}$ denote the accuracy with the trigger. We have:

$$\begin{aligned}
&Acc_{(retrained;w/o;T_{\mathbf{e}})} - Acc_{(retrained;w/;T_{\mathbf{e}})} \\
&= \mathbb{E}_{(\mathbf{c}_i,y_i)\sim\mathcal{D}_{test}}[Acc_{\mathbf{c}_i}] - \mathbb{E}_{(\mathbf{c}_i,y_i)\sim\mathcal{D}_{test}}[Acc_{(\mathbf{c}_i;T_{\mathbf{e}})}] \\
&= \mathbb{E}_{(\mathbf{c}_i,y_i)\sim\mathcal{D}_{test}}[Acc_{\mathbf{c}_i} - Acc_{(\mathbf{c}_i;T_{\mathbf{e}})}] \\
&\leq \mathbb{E}_{(\mathbf{c}_i,y_i)\sim\mathcal{D}_{test}}[\Delta Acc] \\
&= p \cdot \Delta Acc,
\end{aligned}$$

where $p$ is the fraction of data points with the trigger, and $\Delta Acc = \mathbb{E}_{(\mathbf{c}_i,y_i)\sim\mathcal{D}_{test}}[Acc_{(\mathbf{c}_i;T_{\mathbf{e}})} - Acc_{\mathbf{c}_i}]$ is the average decrease in accuracy due to the trigger. This upper bound indicates that the decrease in accuracy is directly proportional to the fraction of data points with the trigger.

In this section, we derive the lower and upper bounds for the success rate of the CAT attack during the Test Time II phase. Let $Acc_{(retrained;w/o;T_{\mathbf{e}})}$ denote the accuracy of the retrained CBM on clean

data without the trigger, and $Acc_{(retrained;w/;T_e)}$ be the accuracy on data with the trigger activated. The success rate of the attack is defined as the proportion of instances originally not labeled as $y_{tc}$ that are misclassified as $y_{tc}$ due to the trigger. Assuming that the trigger is effective, we expect a significant decrease in accuracy when the trigger is present. Let $\Delta Acc$ denote the difference between the accuracy with and without the trigger:

$$\Delta Acc = Acc_{(retrained;w/o;T_e)} - Acc_{(retrained;w/;T_e)}. \tag{10}$$

The lower bound for the success rate can be derived as:

$$\text{Success Rate} \geq \frac{\Delta Acc}{1 - Acc_{(retrained;w/o;T_e)}}. \tag{11}$$

This bound represents the minimum success rate achievable if all the misclassifications due to the trigger are instances originally not labeled as $y_{tc}$. Conversely, the upper bound for the success rate is given by:

$$\text{Success Rate} \leq \frac{\Delta Acc}{Acc_{(retrained;w/o;T_e)}}. \tag{12}$$

This bound assumes that all the misclassifications due to the trigger are from instances originally not labeled as $y_{tc}$, and no clean instances are misclassified. By evaluating the model's performance within these bounds, we can assess the effectiveness of the CAT attack in practice.

## E  LOWER OR UPPER BOUND FOR BAYESIAN CAT

We will employ Bayesian methods to estimate the probability of trigger activated in CAT, and use this to optimize our experimental attempts in further sections. We assume that $\theta$ is the probability of trigger activated in CAT and $\theta \in [0, 1]$. Assuming that we have conducted $N$ backdoor injection experiments on the dataset and the backdoor was triggered $k$ times, where $N$ and $k$ are given. Now we will derive the prior distribution for $\theta$. Clearly, the activation of the trigger will result in one of two states: 1 or 0. The Beta distribution is defined over the interval [0,1] and is a conjugate prior for the binomial distribution, allowing us to obtain a closed-form solution. Therefore, we will use the Beta distribution here, i.e., $\theta \sim \text{Beta}(\alpha, \beta)$. Note that the parameter $\beta$ here is different from the regressor $f$ ones. Then the PDF(prior probability density function) for $\theta$ using the Beta distribution can be expressed as follows:

$$p(\theta) = \frac{\Gamma(\alpha + \beta)}{\Gamma(\alpha)\Gamma(\beta)} \theta^{\alpha-1} (1 - \theta)^{\beta-1}, \ 0 \leq \theta \leq 1, \tag{13}$$

where $\alpha, \beta$ are the prior parameters, $\Gamma(\cdot)$ is the Gamma function.

Now we will establish the likelihood function for the parameter $\theta$. We assume that the probability of triggering a backdoor in each backdoor injection experiment is independent, and through observation, $k$ out of $N$ experiments are successful. The likelihood function for a binomial distribution is:

$$L(\theta) = p(k|\theta) = \binom{N}{k} \theta^k (1 - \theta)^{N-k} \tag{14}$$

According to Bayes' theorem, we obtain:

$$p(\theta|k) = \frac{L(\theta)p(\theta)}{\int_0^1 L(\theta)p(\theta)\, d\theta}, \tag{15}$$

where the denominator is a normalization constant and is independent of $\theta$, we can first calculate the unnormalized posterior distribution and then normalize it by identifying its distribution form.

The unnormalized posterior distribution will satisfy:

$$p(\theta|k) \propto \theta^k (1 - \theta)^{N-k} \cdot \theta^{\alpha-1} (1 - \theta)^{\beta-1}$$
$$= \theta^{\alpha+k-1} (1 - \theta)^{\beta+N-k-1}$$

Same with Beta distribution, this distribution form is identified as:

$$\theta|k \sim \text{Beta}(\alpha', \beta'), \tag{16}$$

where posterior parameters $\alpha' = \alpha + k, \beta' = \beta + N - k$.

Using the posterior distribution we can derive the upper and lower bound of our CAT. Our goal is to obtain the $(1 - \gamma)\%$ confidence interval for $\theta$. Recall the definition of lower or upper bound:

$$p(\theta \leq \theta_{lower}) = \frac{\gamma}{2}, \quad p(\theta \leq \theta_{upper}) = 1 - \frac{\gamma}{2} \tag{17}$$

The bounds for $\theta$ could be expressed below:

$$\theta_{lower} = \text{BetaCDF}^{-1}(\frac{\gamma}{2}, \alpha', \beta') \tag{18}$$

$$\theta_{upper} = \text{BetaCDF}^{-1}(1 - \frac{\gamma}{2}, \alpha', \beta'), \tag{19}$$

where the term $\text{BetaCDF}^{-1}(p, \alpha', \beta')$ represents the $p$-th quantile of the Beta distribution with parameters $\alpha'$ and $\beta'$, and CDF is the cumulative distribution function.

### E.1 PARAMETER ESTIMATION

The PDF of beta distribution is formed as:

$$f(\theta; \alpha, \beta) = \frac{\theta^{\alpha-1}(1 - \theta)^{\beta-1}}{B(\alpha, \beta)}, \tag{20}$$

where term $B(\alpha, \beta)$ is the Beta function. We use MLE (Maximum Likelihood Estimation) to estimate the parameter in beta distribution. Assuming that the observations value of $\theta$ are $\{\theta_1, \theta_2, \cdots, \theta_n\}$ and $\theta_i \in [0, 1]$. The likelihood function of Beta distribution is expressed as:

$$L(\alpha, \beta) = \prod_{i=1}^{n} f(\theta_i, \alpha, \beta), \tag{21}$$

and we transform it into logarithm format:

$$\log L(\alpha, \beta) = \Sigma_{i=1}^{n}[(\alpha - 1)\log(\theta_i) + (\beta - 1)\log(1 - \theta_i)] - n \log B(\alpha, \beta), \tag{22}$$

and the Beta function could be calculated by Gamma function:

$$B(\alpha, \beta) = \frac{\Gamma(\alpha)\Gamma(\beta)}{\Gamma(\alpha + \beta)}.$$

Finally we solve the optimal problems with parameter $\alpha, \beta$ to meet the requirement:

$$\max \log L(\alpha, \beta) = \max \Sigma_{i=1}^{n}[(\alpha - 1)\log(\theta_i) + (\beta - 1)\log(1 - \theta_i)] - n \log B(\alpha, \beta).$$

## F CAT ROBUSTNESS

In our CAT framework, we will not only consider the effectiveness of the attack but also evaluate its robustness against generalized defenses. Models based on random perturbations tend to have strong generalization capabilities, and we will derive the probability of activating the trigger even under random perturbations. Assume that $S \in \mathcal{S}$ is a random perturbation from perturbation space, the definition of CAT Robustness could be expressed as below:

$$R = P_{\mathbf{c},S}\{f(S(\mathbf{c} \oplus \tilde{\mathbf{c}})) = y_{tc}\}, \tag{23}$$

where term $P_{\mathbf{c},S}$ represents the joint probability distribution of $\mathbf{c}$ and $S$. In CAT, concept vector $\mathbf{c}$ and perturbation $S$ are random independent variables. So we could decompose the joint probability distribution into the following expression:

$$P_{\mathbf{c},S} = P_{\mathbf{c}} \cdot P_S, \tag{24}$$

then the CAT robustness could be expressed as follows:

$$R = \int_{\mathcal{C}} \int_{\mathcal{S}} \mathbb{I}\{f(S(\mathbf{c} \oplus \tilde{\mathbf{c}})) = y_{tc}\} dP_{\mathbf{c}}(\mathbf{c}) dP_S(S) \tag{25}$$

where equation 25 is a stochastic differential equation. For a fixed concept vector $\mathbf{c}$, the CAT robustness will be follows:

$$R_{\mathbf{c}} = P_S\{f(S(\mathbf{c} \oplus \tilde{\mathbf{c}})) = y_{tc} | \mathbf{c}\} \tag{26}$$

Therefore, the overall CAT robustness is:

$$R = \mathbb{E}_{\mathbf{c}}[R_{\mathbf{c}}] = \int_{\mathcal{C}} R_{\mathbf{c}} dP_{\mathbf{c}}(\mathbf{c}) \tag{27}$$

## G   THEORETICAL FOUNDATION OF ITERATIVE POISONING

The iterative poisoning strategy in CAT+ is grounded in the concept of maximizing the impact of the backdoor trigger while maintaining stealthiness. To formalize this, we first introduce the concept of *information gain* to quantify the change in the model's understanding of the target class after applying the trigger.

**Information Gain:** The information gain $\mathcal{I}(c_{select}, P_{select})$ is a measure of the additional information the model gains about the target class $y_{tc}$ when the concept $c_{select}$ is perturbed using operation $P_{select}$. It can be defined as the mutual information between the target class and the perturbed concept, given by:

$$\mathcal{I}(c_{select}, P_{select}) = \mathbb{H}(y_{tc}) - \mathbb{H}(y_{tc}|c_{select}, P_{select}), \tag{28}$$

where $\mathbb{H}(y_{tc})$ is the entropy of the target class distribution and $\mathbb{H}(y_{tc}|c_{select}, P_{select})$ is the conditional entropy of the target class given the perturbed concept.

**Optimal Concept Selection:** In each iteration, we aim to maximize the information gain to ensure that the trigger has the most significant impact on the model's prediction. To achieve this, we define the *information gain ratio* as:

$$\mathcal{R}(c_{select}, P_{select}) = \frac{\mathcal{I}(c_{select}, P_{select})}{\mathbb{H}(y_{tc})}, \tag{29}$$

which represents the relative increase in information about the target class due to the perturbation.

**Z-score Revisited:** The Z-score $\mathcal{Z}(c_{select}, P_{select})$ introduced earlier is closely related to the information gain ratio. In fact, we can show that the Z-score is a monotonic function of the information gain ratio, such that a higher Z-score corresponds to a higher information gain ratio. This relationship allows us to use the Z-score as a proxy for selecting the optimal concept and operation in each iteration.

## H   DATASETS

**CUB.** The Caltech-UCSD Birds-200-2011 (CUB)Wah et al. (2011) dataset is designed for bird classification and contains 11,788 bird photographs across 200 species. Additionally, it includes 312 binary bird attributes to represent high-level semantic information. Previous work mostly followed the preprocessing steps outlined by Koh et al. (2020). They first applied majority voting to resolve concept disparities across instances from the same class, then selected attributes that appeared in at least 10 classes, ultimately narrowing the selection to 116 binary attributes. However, in our experiments, we preprocess the data slightly differently. We do not attempt to eliminate the disparity across instances from the same class, meaning we accept that instances from the same species may have different concept representations. Additionally, we select high-frequency attributes at the instance level—specifically, we only use attributes that appear in at least 500 instances. Finally, we retain 116 attributes, with over 90% overlap with the attributes selected by Koh et al. (2020).

**AwA.** The Animals with Attributes (AwA) Xian et al. (2018) dataset contains 37,322 images across 50 animal categories, with each image annotated with 85 binary attributes. We split the images equally by class into training and test datasets, resulting in 18,652 images in the training set and 18,670 images in the test set. There is no modification for the binary attributes.

## I   EXPERIMENT SETTINGS

We conducted all of our experiments on a NVIDIA A40 GPU. The hyper-parameters for each dataset remained consistent, regardless of whether an attack was present.

During training, we use a batch size of 64 and a learning rate of 1e-4. The Adam optimizer is applied with a weight decay of 5e-5, alongside an exponential learning scheduler with $\gamma = 0.95$. The concept loss weight $\lambda$ is set to 0.5. For image augmentations, we follow the approach of

Koh et al. (2020) with a slight modification in resolution. Each training image is augmented using random color jittering, random horizontal flips, and random cropping to a resolution of 256. During inference, the original image is center-cropped and resized to 256. For AwA dataset, We use a batch size of 128, while all other hyper-parameters and image augmentations remain consistent with those used for the CUB dataset.

## J    EXTEND EXPERIMENT RESULTS

In this section, we give more detailed experiment results, and we also conducted our attack experiments across different target classes. Table 2 and Table 3 gives the detailed experiment results for CUB dataset when the target class is set to 0, we see that CAT+ can achieve high ASR, while maintain a high performance on clean data, showing the effectiveness of CAT+, even with a small injection rate (1%). Table 4 and Table 5 shows the detailed experiment results for AwA dataset when the target class is set to 0, while Table 6 and Table 7 shows the detailed experiment results for AwA dataset when the target class is set to 2. These results shows that our CAT and CAT+ works well across different target classes. Table 8 gives the results on CUB dataset with a fixed trigger size and injection rate, which shows some performance disparity across different target classes.

|     | 1% | | 2% | | 3% | | 5% | | 7% | | 10% | |
| --- | --- | --- | --- | --- | --- | --- | --- | --- | --- | --- | --- | --- |
|     | CAT | CAT+ | CAT | CAT+ | CAT | CAT+ | CAT | CAT+ | CAT | CAT+ | CAT | CAT+ |
| 2  | 0.52 | 3.85 | 1.27 | 8.50 | 1.94 | 9.58 | 2.85 | 14.28 | 4.49 | 20.44 | 7.55 | 26.65 |
| 5  | 0.23 | 12.11 | 0.92 | 19.99 | 1.51 | 24.58 | 2.01 | 28.40 | 4.37 | 40.42 | 7.15 | 42.45 |
| 8  | 2.06 | 23.09 | 5.12 | 36.80 | 5.97 | 43.15 | 7.46 | 44.26 | 14.69 | 55.38 | 19.17 | 63.72 |
| 10 | 0.90 | 18.06 | 3.57 | 26.63 | 6.11 | 42.21 | 8.34 | 47.33 | 9.39 | 52.38 | 16.20 | 50.78 |
| 12 | 3.85 | 19.08 | 11.54 | 37.23 | 18.20 | 49.84 | 17.80 | 51.58 | 28.19 | 56.52 | 36.26 | 64.40 |
| 15 | 3.92 | 14.10 | 13.58 | 24.08 | 16.78 | 31.80 | 22.95 | 32.43 | 29.56 | 38.67 | 37.46 | 39.37 |
| 17 | 8.26 | 22.80 | 23.18 | 35.69 | 29.27 | 47.76 | 39.56 | 54.70 | 48.32 | 63.60 | 52.64 | 66.69 |
| 20 | 6.89 | 57.70 | 24.36 | 77.65 | 30.99 | 84.47 | 35.90 | 87.51 | 48.77 | 90.16 | 59.28 | 93.01 |
| 23 | 6.04 | 74.76 | 18.95 | 84.00 | 25.45 | 87.82 | 26.32 | 91.92 | 43.36 | 87.20 | 52.41 | 90.65 |

Table 2: ASR(%) under different injection rates (1% – 10%) and trigger size (2 – 23) in CUB dataset, target class 0.

|     | 1% | | 2% | | 3% | | 5% | | 7% | | 10% | |
| --- | --- | --- | --- | --- | --- | --- | --- | --- | --- | --- | --- | --- |
|     | CAT | CAT+ | CAT | CAT+ | CAT | CAT+ | CAT | CAT+ | CAT | CAT+ | CAT | CAT+ |
| 2  | 80.58 | 80.70 | 80.08 | 80.05 | 79.08 | 79.72 | 78.75 | 78.51 | 76.99 | 77.55 | 74.56 | 74.27 |
| 5  | 80.81 | 80.00 | 79.92 | 79.72 | 80.12 | 79.50 | 78.44 | 78.46 | 76.94 | 76.46 | 78.92 | 74.68 |
| 8  | 80.95 | 80.74 | 79.74 | 79.75 | 79.50 | 79.72 | 77.94 | 78.49 | 77.11 | 77.55 | 74.84 | 75.13 |
| 10 | 80.70 | 80.43 | 80.27 | 80.19 | 79.51 | 79.38 | 78.31 | 78.31 | 77.34 | 77.06 | 74.82 | 72.95 |
| 12 | 81.08 | 80.62 | 79.93 | 79.82 | 79.03 | 79.22 | 78.55 | 77.67 | 77.42 | 76.91 | 74.99 | 74.85 |
| 15 | 80.72 | 80.24 | 80.19 | 79.88 | 80.00 | 78.68 | 78.34 | 77.49 | 77.46 | 77.08 | 74.96 | 74.91 |
| 17 | 80.84 | 80.51 | 79.82 | 79.96 | 79.20 | 79.58 | 78.31 | 78.29 | 76.84 | 77.32 | 75.53 | 74.91 |
| 20 | 80.82 | 80.81 | 80.39 | 80.26 | 79.70 | 79.93 | 78.22 | 78.82 | 77.46 | 77.49 | 75.03 | 75.53 |
| 23 | 80.57 | 80.19 | 80.31 | 80.43 | 79.58 | 79.62 | 78.72 | 78.10 | 77.72 | 76.96 | 75.42 | 74.83 |

Table 3: Task Accuracy under different injection rates (1% – 10%) and trigger size (2 – 23) of test in CUB dataset, target class 0. Task Accuracy shows the effectiveness of mapping $f$ in our CATs. The original task accuracy in CUB is $80.70\%$.

## K    MORE EXPERIMENTS

### K.1    EXPERIMENTS ON DIFFERENT BACKBONES

We evaluate the attack performance by using another pre-trained vision backbone, Vision Transformer (VIT)Dosovitskiy (2020), we resize the input images to $224 \times 224$ to fit the input dimension

|  | 2% | | 5% | | 10% | |
|---|---|---|---|---|---|---|
|  | CAT | CAT+ | CAT | CAT+ | CAT | CAT+ |
| 2 | 3.14 | 0.50 | 7.31 | 1.54 | 15.18 | 4.29 |
| 5 | 5.87 | 0.48 | 13.71 | 1.51 | 27.67 | 5.39 |
| 8 | 9.22 | 1.74 | 23.35 | 5.95 | 41.17 | 12.02 |
| 10 | 14.26 | 4.48 | 26.61 | 29.70 | 44.57 | 47.20 |
| 12 | 24.96 | 1.13 | 41.11 | 41.49 | 55.17 | 59.95 |
| 15 | 26.89 | 2.53 | 42.30 | 5.77 | 59.49 | 62.74 |
| 17 | 31.43 | 25.32 | 50.03 | 45.37 | 64.80 | 65.32 |

Table 4: ASR (%) under different injection rates (2%, 5%, 10%) and trigger size ($2 - 17$) in AwA dataset, target class 0.

|  | 2% | | 5% | | 10% | |
|---|---|---|---|---|---|---|
|  | CAT | CAT+ | CAT | CAT+ | CAT | CAT+ |
| 2 | 83.06 | 83.09 | 81.20 | 81.29 | 78.48 | 77.04 |
| 5 | 83.12 | 83.18 | 80.88 | 81.20 | 76.59 | 76.53 |
| 8 | 83.02 | 83.19 | 80.67 | 80.87 | 76.35 | 76.85 |
| 10 | 83.23 | 83.20 | 80.86 | 81.11 | 78.64 | 72.45 |
| 12 | 83.36 | 82.79 | 80.70 | 80.76 | 76.90 | 76.68 |
| 15 | 83.37 | 83.37 | 80.86 | 80.93 | 76.40 | 76.56 |
| 17 | 83.00 | 82.87 | 80.87 | 80.62 | 76.13 | 76.99 |

Table 5: Task Accuracy under different injection rates (2%, 5%, 10%) and trigger size ($2 - 17$) in AwA dataset, target class 0. Task Accuracy shows the effectiveness of mapping $f$ in our CATs. The original task accuracy for AwA is $84.68\%$.

of VIT, the results are shown in Table 10. When the pre-trained backbone comes to VIT, our CAT+ still keeps both the stealthiness and high ASR: The lowest task accuracy with no trigger activated is $81.48\%$, which guarantees the stealthiness of our attack, and highest ASR comes to $72.05\%$. This experiment claims that our CAT+ adapts different pre-trained backbone and portability.

## K.2 EXPERIMENTS ON MORE DATASETS

We also evaluate our attack performance on Large-scale Attribute Dataset (LAD) Zhao et al. (2019), which contains 78,017 images in total. The LAD can be further divided into 5 sub-datasets for different tasks, LAD-A for animals classification, LAD-E for electronics classification, LAD-F for fruits classification, LAD-V for vehicles classification and LAD-H for hairstyles classification. The statistics of the five sub-datasets are summarized in Table 11. For each class in LAD, there are 20 images labeled with binary attributes, while the remaining images are unlabeled with attributes, to handle this, we labeled the attributes for those attribute-unlabeled images by:

$$c_{ij}^{\mathcal{A}} = \mathbb{I}(p < \overline{c_{ij}^{A}}), \tag{30}$$

where $c_{ij}^{A}$ is the $j$-th concept of class $i$ for dataset $\mathcal{A}$, $\overline{c_{ij}^{A}} = \frac{1}{n_i} \sum c_{ij}^{A}$ is the average value of this concept, $n_i$ refers to the number of attribute-labeled images, and $p$ is a random variable sampled from a uniform distribution on the interval $[0, 1]$.

Then we follow the same experiment settings in I and evaluate the attack performance in each sub-datasets except for LAD-H, for the original accuracy of LAD-H is not ideal. The original accuracies for each sub-datasets are shown in Table 11. We evaluate the performances of CAT and CAT+ on LAD-A and LAD-E, the results are shown in Table 12 13, 14, 15, and we evaluate the performance of CAT on LAD-F and CAT+ on LAD-V, the results are shown in 16, 17. By evaluating each sub-dataset, we observed a significant increase in attack success rates (ASR) as the injection rate increased, particularly with the CAT+ method, which demonstrated more pronounced effectiveness compared to the CAT method.

|    | 2% | | 5% | | 10% | |
|----|------|------|-------|-------|-------|-------|
|    | CAT | CAT+ | CAT | CAT+ | CAT | CAT+ |
| 2  | 2.27 | 2.09 | 3.18 | 4.55 | 10.35 | 13.46 |
| 5  | 5.40 | 5.33 | 12.25 | 12.28 | 22.03 | 24.31 |
| 8  | 13.13 | 13.69 | 20.99 | 24.59 | 39.67 | 44.93 |
| 10 | 14.32 | 12.76 | 25.17 | 27.02 | 43.79 | 47.56 |
| 12 | 21.22 | 19.68 | 29.12 | 51.31 | 49.15 | 58.78 |
| 15 | 34.95 | 40.08 | 51.45 | 56.88 | 74.22 | 75.21 |
| 17 | 44.34 | 44.18 | 62.42 | 64.85 | 79.22 | 79.13 |

Table 6: ASR (%) under different injection rates (2% – 10%) and trigger size (2 – 17) in AwA dataset, target class 2.

|    | 2% | | 5% | | 10% | |
|----|------|------|-------|-------|-------|-------|
|    | CAT | CAT+ | CAT | CAT+ | CAT | CAT+ |
| 2  | 82.79 | 82.76 | 81.51 | 79.86 | 76.89 | 74.44 |
| 5  | 82.91 | 82.81 | 81.26 | 78.81 | 76.83 | 74.49 |
| 8  | 83.00 | 82.64 | 81.47 | 80.03 | 77.03 | 75.11 |
| 10 | 83.42 | 82.59 | 81.81 | 79.59 | 77.16 | 73.75 |
| 12 | 83.20 | 82.21 | 81.03 | 74.07 | 77.41 | 74.32 |
| 15 | 83.15 | 82.28 | 81.42 | 79.35 | 76.61 | 74.86 |
| 17 | 83.26 | 82.51 | 81.01 | 78.65 | 76.61 | 74.32 |

Table 7: Task Accuracy (%) under different injection rates (2% – 10%) and trigger size (2 – 17) in AwA dataset, target class 2. The original task accuracy for AwA is $84.68\%$.

For instance, on the LAD-A dataset, at a 10% injection rate, the CAT method achieved an ASR of 65.87%, while the CAT+ method reached an ASR of 93.82%. This difference highlights the enhanced efficiency of the CAT+ method, which benefits from its iterative poisoning strategy and the subtle selection of concepts for modification. Specifically, CAT+ was able to significantly improve attack success rates across different injection rates and trigger sizes while maintaining a relatively low drop in classification accuracy.

On the LAD-E dataset, where the original accuracy was 77.82%, both CAT and CAT+ showed similar trends. Despite the increased challenge posed by the LAD-E dataset, the CAT+ method still achieved a high ASR, particularly at a 10% injection rate and larger trigger sizes. In this case, CAT+ achieved an ASR of 84.56%, compared to 50.33% for the CAT method, further underscoring the superior performance of CAT+.

For the LAD-F dataset, with an original accuracy of 89.59%, we found that even at low injection rates (such as 2%), the CAT+ method exhibited a high ASR, reaching as high as 96.69%. This result further validates the broad applicability and strong effectiveness of the CAT+ method across various tasks and datasets.

On the LAD-V dataset, the performance of both the CAT and CAT+ methods followed a similar pattern, but the CAT+ method consistently achieved higher ASR, particularly at higher injection rates, where the ASR reached over 80%. This indicates that CAT+ performs especially well on this dataset.

Overall, the CAT+ method consistently demonstrated a significant advantage in attack success rate across most sub-datasets. These results confirm the superior effectiveness of the CAT+ method, especially in multi-task learning scenarios with high injection rates and larger trigger sizes. In comparison, although the CAT method also achieved relatively high attack success rates in some cases, the CAT+ method's overall superiority across multiple datasets was more pronounced.

| | Task Accuracy(%) | | ASR(%) | |
|---|---|---|---|---|
| Target Class | CAT | CAT+ | CAT | CAT+ |
| 0 | 75.03 | 75.53 | 59.28 | 93.01 |
| 4 | 74.54 | 74.73 | 1.85 | 0.10 |
| 8 | 75.06 | 75.56 | 74.24 | 52.63 |
| 12 | 74.94 | 75.72 | 53.38 | 1.08 |
| 16 | 75.16 | 75.72 | 40.91 | 68.81 |
| 20 | 75.34 | 75.58 | 1.28 | 12.02 |
| 24 | 74.37 | 74.91 | 0.52 | 54.48 |
| 28 | 74.27 | 74.65 | 35.48 | 17.87 |
| 32 | 74.70 | 75.58 | 37.68 | 2.32 |
| 36 | 74.96 | 75.27 | 37.99 | 6.50 |
| 40 | 74.46 | 74.96 | 42.35 | 10.40 |
| 44 | 74.89 | 75.22 | 17.10 | 23.77 |
| 48 | 75.09 | 75.73 | 49.77 | 4.51 |
| 52 | 74.68 | 74.97 | 82.15 | 95.11 |
| 56 | 75.22 | 75.23 | 70.99 | 57.36 |
| 60 | 74.97 | 75.37 | 2.60 | 24.01 |
| 64 | 74.85 | 74.58 | 43.63 | 84.27 |
| 68 | 75.09 | 75.58 | 39.30 | 9.63 |
| 72 | 74.99 | 75.34 | 47.99 | 59.06 |
| 76 | 74.53 | 74.06 | 46.16 | 30.55 |
| 80 | 75.03 | 75.61 | 62.51 | 3.71 |
| 84 | 74.49 | 74.87 | 9.82 | 75.83 |
| 88 | 74.75 | 74.66 | 51.13 | 32.44 |
| 92 | 75.34 | 75.60 | 39.68 | 72.71 |
| 96 | 74.82 | 75.20 | 17.73 | 11.64 |

Table 8: Task Accuracy and ASR for different Target Classes from 0 to 196. The test dataset is CUB, trigger size is 20 and the injection rate is 10% (1 of 2).

| | Task Accuracy(%) | | ASR(%) | |
|---|---|---|---|---|
| Target Class | CAT | CAT+ | CAT | CAT+ |
| 100 | 74.94 | 75.42 | 48.96 | 62.16 |
| 104 | 74.84 | 74.92 | 1.07 | 0.07 |
| 108 | 74.49 | 74.63 | 53.02 | 11.00 |
| 112 | 73.92 | 74.46 | 10.76 | 40.42 |
| 116 | 75.35 | 75.51 | 11.50 | 18.83 |
| 120 | 74.58 | 74.59 | 11.49 | 12.58 |
| 124 | 74.97 | 75.70 | 0.24 | 61.04 |
| 128 | 74.66 | 75.15 | 10.15 | 54.34 |
| 132 | 74.99 | 75.66 | 9.54 | 14.23 |
| 136 | 74.73 | 74.99 | 2.95 | 0.83 |
| 140 | 74.58 | 74.97 | 7.86 | 64.01 |
| 144 | 75.28 | 75.35 | 90.61 | 9.94 |
| 148 | 74.78 | 75.42 | 0.36 | 76.79 |
| 152 | 75.09 | 75.22 | 80.46 | 17.19 |
| 156 | 74.54 | 75.94 | 1.37 | 35.98 |
| 160 | 74.99 | 75.16 | 1.80 | 20.07 |
| 164 | 74.85 | 75.56 | 13.78 | 6.61 |
| 168 | 75.11 | 75.83 | 50.44 | 50.70 |
| 172 | 74.70 | 74.73 | 29.96 | 66.69 |
| 176 | 74.77 | 74.92 | 2.29 | 0.03 |
| 180 | 74.91 | 74.58 | 4.35 | 76.96 |
| 184 | 75.16 | 75.66 | 27.93 | 5.95 |
| 188 | 74.44 | 75.73 | 57.41 | 21.74 |
| 192 | 74.25 | 75.75 | 41.39 | 46.30 |
| 196 | 74.73 | 75.80 | 23.72 | 20.96 |

Table 9: Task Accuracy and ASR for different Target Classes from 0 to 196. The test dataset is CUB, trigger size is 20 and the injection rate is 10% (2 of 2).

| | 2% | | 5% | | 10% | |
|---|---|---|---|---|---|---|
| | ACC(%) | ASR(%) | ACC(%) | ASR(%) | ACC(%) | ASR(%) |
| 8 | 86.54 | 9.89 | 85.42 | 16.12 | 83.02 | 20.23 |
| 10 | 86.24 | 7.41 | 85.23 | 13.60 | 82.79 | 22.28 |
| 12 | 86.57 | 11.31 | 85.52 | 21.56 | 83.03 | 31.75 |
| 15 | 86.76 | 20.96 | 85.26 | 30.76 | 82.62 | 42.30 |
| 17 | 86.66 | 43.86 | 85.31 | 58.48 | 82.10 | 70.85 |
| 20 | 86.40 | 48.47 | 84.62 | 60.27 | **81.48** | **72.05** |

Table 10: Task Accuracy(%) and ASR(%) under different injection rates(2% - 10%) and trigger size in CUB dataset, target class 0, vision backbone is a pretrained VIT, the attack mode is fixed to CAT+, the Original Accuracy is 87.30%.

## L HUMAN EVALUATION DETAILS

### L.1 HUMAN EVALUATION PROTOCOL

The human evaluators were provided with the following instructions:

1. **Dataset Description:** You will be presented with a dataset consisting of 60 concept representations, each associated with an input sample (x) and its corresponding class labels (c, y). Among these, 30 concept representations have been backdoored using a concept-based trigger, while the remaining 30 are clean.

| | Training Size | Test Size | # of Concept | # of Class | Original ACC(%) |
|---|---|---|---|---|---|
| LAD-A | 9280 | 3960 | 123 | 50 | 88.54 |
| LAD-E | 12916 | 5555 | 75 | 50 | 77.82 |
| LAD-F | 13606 | 5850 | 58 | 50 | 89.59 |
| LAD-V | 11979 | 5101 | 81 | 50 | 84.30 |
| LAD-H | 6829 | 2941 | 22 | 30 | 58.25 |

Table 11: Statistics and Original Task Accuracy(%)of each LAD sub-datasets

| | 2% | | 5% | | 10% | |
|---|---|---|---|---|---|---|
| | ACC(%) | ASR(%) | ACC(%) | ASR(%) | ACC(%) | ASR(%) |
| 2 | 87.95 | 8.13 | 85.68 | 10.35 | 81.94 | 21.30 |
| 5 | 87.65 | 33.04 | 85.81 | 30.14 | 81.67 | 33.40 |
| 8 | 87.80 | 55.51 | 85.93 | 57.29 | 81.84 | 63.09 |
| 10 | 87.78 | 61.42 | 85.96 | 60.51 | 82.42 | 65.87 |
| 12 | 87.58 | 50.91 | 85.76 | 54.47 | 81.84 | 62.26 |
| 15 | 87.47 | 55.54 | 85.76 | 60.12 | 82.40 | 61.58 |
| 17 | 88.16 | 60.25 | 86.09 | 61.32 | 82.42 | 65.21 |
| 20 | 87.95 | 68.66 | 85.73 | 74.33 | 81.92 | 14.45 |

Table 12: Task Accuracy(%) and ASR(%) under different injection rates(2% - 10%) and trigger size (2-20) in LAD-A dataset, target class 0, the attack mode is CAT, the Original Accuracy is 88.54%.

2. **Task:** Your task is to identify which data has been backdoored.

3. **Evaluation Criteria:** Analyze the concept space for any subtle modifications that might indicate the presence of a backdoor trigger. Avoid relying on the input samples or class labels.

## L.2 POST-EVALUATION INTERVIEWS AND INSIGHTS

After completing the evaluation, the evaluators were interviewed to gather their thoughts and insights on the task. The interview questions included:

1. **Q1:** Describe your approach to distinguishing between backdoored and clean concept representations.

2. **Q2:** Did you notice any specific patterns or changes in the concept space that helped you identify the backdoor samples?

3. **Q3:** How difficult was it to identify the backdoor attacks compared to your initial expectations?

4. **Q4:** What factors do you think contributed to the difficulty in detecting the backdoor in the concept space?

**Evaluator 1:**

1. **A1:** I focused on the relationships between concepts and images.

2. **A2:** The trigger concepts seemed to have a more pronounced effect, but no consistent pattern was evident.

3. **A3:** It was much more challenging than expected due to the subtlety of the changes.

4. **A4:** There were so many concepts that I got distracted, and a lot of datasets were actually mislabeled. It felt like looking for a needle in a haystack. It was so painful. **It's like pouring Coca-Cola into Pepsi.**

**Evaluator 2:**

1. **A1:** I looked for inconsistencies or anomalies that didn't align with the expected concept representation.

|    | 2% | | 5% | | 10% | |
|----|--------|--------|--------|--------|--------|--------|
|    | ACC(%) | ASR(%) | ACC(%) | ASR(%) | ACC(%) | ASR(%) |
| 2  | 87.22 | 14.53 | 85.45 | 19.76 | 83.41 | 32.54 |
| 5  | 88.16 | 12.86 | 86.46 | 31.23 | 83.16 | 37.69 |
| 8  | 88.38 | 37.45 | 85.66 | 43.26 | 83.06 | 52.48 |
| 10 | 87.47 | 35.26 | 86.16 | 48.38 | 82.78 | 64.72 |
| 12 | 87.65 | 53.92 | 85.48 | 69.29 | 81.92 | 69.24 |
| 15 | 87.90 | 64.56 | 86.16 | 67.69 | 82.42 | 79.67 |
| 17 | 87.73 | 45.19 | 85.58 | 63.07 | 83.08 | 74.93 |
| 20 | 87.20 | 53.48 | 86.36 | 77.26 | 82.85 | 77.03 |

Table 13: Task Accuracy(%) and ASR(%) under different injection rates(2% - 10%) and trigger size (2 - 20) in LAD-A dataset, target class 0, the attack mode is CAT+, the Original Accuracy is 88.54%.

|    | 2% | | 5% | | 10% | |
|----|--------|--------|--------|--------|--------|--------|
|    | ACC(%) | ASR(%) | ACC(%) | ASR(%) | ACC(%) | ASR(%) |
| 2  | 76.24 | 7.70  | 74.62 | 16.59 | 71.05 | 35.22 |
| 5  | 76.33 | 16.52 | 74.60 | 30.39 | 70.62 | 50.33 |
| 8  | 76.62 | 36.96 | 75.03 | 59.33 | 74.23 | 73.00 |
| 10 | 76.42 | 55.54 | 74.60 | 71.26 | 71.07 | 78.63 |
| 12 | 76.47 | 42.56 | 74.51 | 70.96 | 71.11 | 81.76 |
| 15 | 76.74 | 51.69 | 74.58 | 69.57 | 70.98 | 75.19 |

Table 14: Task Accuracy(%) and ASR(%) under different injection rates(2% - 10%) and trigger size (2 - 15) in LAD-E dataset, target class 0, the attack mode is CAT, the Original Accuracy is 77.82%.

2. **A2:** There were slight shifts in emphasis on certain concepts, but no clear pattern.

3. **A3:** It was harder than anticipated due to the almost imperceptible changes.

4. **A4:** The subtlety of the trigger and the high dimensionality of the concept space made it challenging.

**Evaluator 3:**

1. **A1:** I searched for deviations from expected concept co-occurrence patterns and structure.

2. **A2:** Minor disruptions in co-occurrence patterns were observed, but not consistent enough.

3. **A3:** It was significantly more difficult than expected due to the stealthiness of the changes.

4. **A4:** The trigger's stealthiness and the complexity of the concept space made detection difficult.

## M  LLM EVALUATION DETAILS

The LLM evaluation protocol involved the following steps:

1. **Dataset Preparation:** The 60 concept representations (30 backdoor-attacked and 30 clean), along with their corresponding input samples (x) and class labels (c, y), were provided as input to GPT4-Vision.

2. **Prompt Design:** The prompt for GPT4-Vision was, "image This is the concept of this image what i give: concept: weight, determine if it has been poisioneed. If poisioned, output 1; otherwise, output 0."

## N  CAT+ CONTINUOUSLY EXTENSION

Following the same notation from CAT+ in previous discussion, and assuming that there are $c_{num}$ types of values in the concept, the continuously CAT+ function $\mathcal{Z}(\cdot)$ is defined as follows:

| | 2% | | 5% | | 10% | |
|---|---|---|---|---|---|---|
| | ACC(%) | ASR(%) | ACC(%) | ASR(%) | ACC(%) | ASR(%) |
| 2 | 83.96 | 3.86 | 74.42 | 37.39 | 71.97 | 57.91 |
| 5 | 76.18 | 26.17 | 74.83 | 40.31 | 71.40 | 56.91 |
| 8 | 76.56 | 52.43 | 74.96 | 65.31 | 72.06 | 75.52 |
| 10 | 76.76 | 29.67 | 74.91 | 46.50 | 71.76 | 70.85 |
| 12 | 76.67 | 22.76 | 74.96 | 37.33 | 71.52 | 45.07 |
| 15 | 77.19 | 12.67 | 74.80 | 28.31 | 72.33 | 35.93 |

Table 15: Task Accuracy(%) and ASR(%) under different injection rates(2% - 10%) and trigger size (2 - 15) in LAD-E dataset, target class 0, the attack mode is CAT+, the Original Accuracy is 77.82%.

| | 2% | | 5% | | 10% | |
|---|---|---|---|---|---|---|
| | ACC(%) | ASR(%) | ACC(%) | ASR(%) | ACC(%) | ASR(%) |
| 2 | 87.86 | 23.95 | 85.40 | 33.74 | 82.72 | 47.41 |
| 5 | 87.95 | 75.89 | 85.42 | 77.59 | 82.58 | 84.56 |
| 8 | 87.40 | 94.61 | 85.62 | 97.22 | 82.84 | 96.69 |
| 10 | 87.90 | 90.04 | 85.81 | 93.80 | 82.60 | 93.82 |

Table 16: Task Accuracy(%) and ASR(%) under different injection rates(2% - 10%) and trigger size (2 - 10) in LAD-F dataset, target class 0, the attack mode is CAT, the Original Accuracy is 89.59%.

(i) Let $n$ be the total number of training samples, and $n_{target}$ be the number of samples from the target class. The initial probability of the target class is $p_0 = n_{target}/n$.

(ii) Given a modified dataset $c_a = \mathcal{D}; c_{select}; P_{select}$, we calculate the conditional probability of the target class given $c_a$ as $p^{(target|c_a)} = \mathbb{H}(target(c_a))/\mathbb{H}(c_a)$, where $\mathbb{H}$ is a function that computes the overall distribution of labels in the dataset.

(iii) Calculate each concept distance $\mathcal{Z}_{c_{select}}$ in selected concept:

$$\mathcal{Z}_{c_{select}} = \Sigma_{i=0}^{i=c_{num}}(c_i - c_{select})^2 \tag{31}$$

(iv) The Z-score for $c_a$ is defined as:

$$\mathcal{Z}(c_a) = \mathcal{Z}_{c_{select}}\mathcal{Z}(c_{select}, P_{select}) = \mathcal{Z}_{c_{select}}\left[p^{(target|c_a)} - p_0\right] / \left[\frac{p_0(1-p_0)}{p^{(target|c_a)}}\right] \tag{32}$$

## O  PRELIMINARY OF DEFENSE AND RESULTS

### O.1  ANALYSIS OF NEURAL CLEANSE DEFENSE AGAINST CAT

In this study, we evaluated the effectiveness of Neural Cleanse in defending against CAT attacks. The backdoored model was configured with the following parameters: CAT Attack, injection rate = 0.1, trigger size = 20, dataset = CUB, and target class = 0. Neural Cleanse is a technique designed to detect and mitigate backdoor attacks in deep learning models by analyzing the model's behavior on specific inputs and identifying potential backdoors.

We applied Neural Cleanse to import the backdoored model and clean test data. Through reverse engineering, we generated trigger and mask images for all categories. Subsequently, we calculated the L1-norm of the reverse-engineered triggers and used this to compute the median, Median Absolute Deviation (MAD), and anomaly index for each category. The anomaly index helps identify potential backdoor target classes by flagging categories with high deviations from the norm.

Due to the large number of classes (200), we present a subset of the results in Table 19.

Neural Cleanse flagged labels 44 and 16 as anomalies due to their anomaly indices exceeding 2.0. However, this does not align with the true target class, which is label 0. For further analysis, the

| | 2% | | 5% | | 10% | |
|---|---|---|---|---|---|---|
| | ACC(%) | ASR(%) | ACC(%) | ASR(%) | ACC(%) | ASR(%) |
| 2 | 83.96 | 3.86 | 82.18 | 7.33 | 79.59 | 13.69 |
| 5 | 83.65 | 11.23 | 81.45 | 22.14 | 78.06 | 32.13 |
| 8 | 83.30 | 55.32 | 81.20 | 67.65 | 78.69 | 81.24 |
| 10 | 83.91 | 64.13 | 82.87 | 79.68 | 79.40 | 80.14 |
| 12 | 83.96 | 75.48 | 82.42 | 83.59 | 78.71 | 90.57 |
| 15 | 83.98 | 72.30 | 81.63 | 81.59 | 79.14 | 85.43 |

Table 17: Task Accuracy(%) and ASR(%) under different injection rates(2% - 10%) and trigger size (2 - 15) in LAD-V dataset, target class 0, the attack mode is CAT+, the Original Accuracy is 84.30%.

| Model | Accuracy | Precision | Recall | F1 Score |
|---|---|---|---|---|
| Human-1 | 0.517 | 0.508 | 1.000 | 0.674 |
| Human-2 | 0.483 | 0.471 | 0.267 | 0.340 |
| Human-3 | 0.483 | 0.333 | 0.033 | 0.061 |
| GPT4v-1 | 0.433 | 0.464 | 0.867 | 0.605 |
| GPT4v-2 | 0.467 | 0.483 | 0.933 | 0.636 |
| GPT4v-3 | 0.483 | 0.492 | 0.967 | 0.652 |

Table 18: Classification Metrics Comparison

| Label | L1-norm | Anomaly Index |
|---|---|---|
| 0 | 25232.596 | 1.038 |
| 1 | 25148.714 | 0.558 |
| 2 | 25489.729 | 2.509 |
| 3 | 25261.745 | 1.205 |
| 5 | 25106.957 | 0.319 |
| 6 | 25182.588 | 0.752 |
| 7 | 24880.898 | 0.975 |
| 8 | 25053.514 | 0.013 |
| 9 | 24892.678 | 0.907 |
| 10 | 24867.008 | 1.054 |
| 11 | 24980.345 | 0.405 |
| 12 | 25042.376 | 0.051 |
| 13 | 25349.604 | 1.707 |
| 16 | 24658.980 | 2.244 |
| 44 | 24596.369 | **2.602** |

Table 19: Anomaly Index for Selected Categories.

mask images, and pattern images obtained through reverse engineering for categories 0, 16, and 44 are visualized in Figure 6.

This failure can be attributed to the unique characteristics of CAT attacks. Unlike conventional backdoor attacks that manipulate inputs, outputs, or model structures, CAT targets the concept layer during training. This attack exploits the reliance of CBMs on interpretable representations, making it distinct and challenging to detect using methods designed for more straightforward backdoor attacks. The attacker has access to the training data but lacks direct control over the concept space during inference, leading to subtle and stealthy manipulations that are not easily observable through standard analysis techniques.

These results suggest that Neural Cleanse may not be effective in defending against CAT attacks. This highlights the need for developing specialized defense mechanisms that can address the unique characteristics of concept-level backdoor attacks. Further research is required to better protect CBMs from such sophisticated threats.

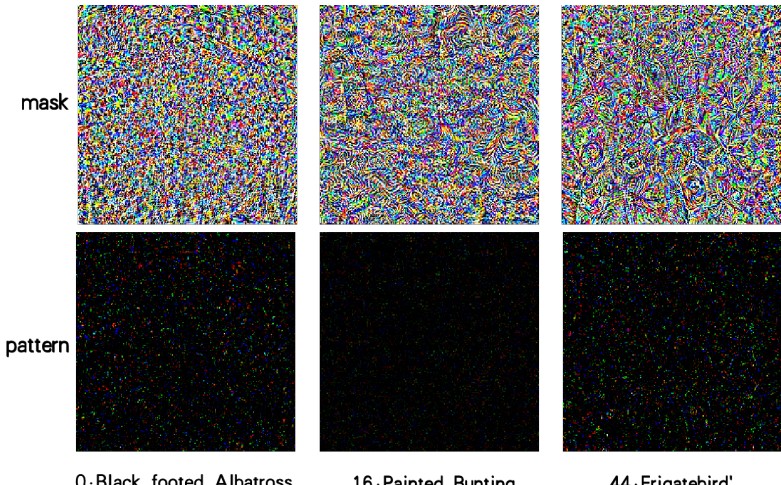

Figure 6: Visualization of Mask Images, and Pattern Images for Classes 0, 16, and 44.

## O.2 DEMO ABOUT A DEFENSE METHOD WHICH WE DESIGN

Given an training dataset $\mathcal{D} = \{(\mathbf{x}_1, \mathbf{c}_1, y_1), (\mathbf{x}_2, \mathbf{c}_2, y_2), \cdots, (\mathbf{x}_n, \mathbf{c}_n, y_n)\}$, where $n$ is the number of data. For concept vectors $\mathbf{c}$, we first encode them from text form into embedding, then use clustering algorithm to cluster them into $m$ groups $\mathcal{F}^j(\mathbf{c}_i)$. Then we divide training dataset into groups following the index of $\mathbf{c}_i$ to generate $m$ sub-datasets. After preparation for the data, we individually train our model upon every sub-dataset and acquire sub-classifier $f^j$. In testing time, every input concept vector divided by the same clustering method into groups and be predicted as a result. At last, the ensemble result is given by the majority vote through $m$ sub-classifiers. Table 20 shows the result of our prototype.

| Clustering Num | Original | CAT | CAT+ | ASR(CAT) | ASR(CAT+) |
|---|---|---|---|---|---|
| Clustering Num 3 | 83.09 | 77.79 | 77.17 | 30.78↓ | 42.75↓ |
| Clustering Num 4 | 83.03 | 78.75 | 78.56 | 11.55↓ | 17.16↓ |
| Clustering Num 5 | 84.24 | 79.51 | 80.76 | 25.95↓ | 16.64↓ |
| Clustering Num 6 | 84.12 | 80.43 | 80.43 | 23.84↓ | 20.12↓ |

Table 20: The Accuracy (%) for each guard model on clean test data for CUB dataset, the Clustering Num denotes a parameter we propose to use in our futher defense framework, the Original denotes to the accuracy when there is no attack. The CAT and CAT+ value refers to the accuracy of defense model and the ASR refers to attack success rate in different models. The experiment settings: injection rate is 5%, trigger size is 20, and the original ASR are 44.66% and 89.68% of CAT and CAT+, respectively.

## P THREAT MODEL

In an image classification task within Concept Bottleneck Models, let the dataset $\mathcal{D}$ comprise $n$ samples, expressed as $\mathcal{D} = \{(\mathbf{x}_i, \mathbf{c}_i, y_i)\}_{i=1}^n$, where $\mathbf{c}_i \in \mathbb{R}^L$ represents the concept vector associated with the input $\mathbf{x}_i$, and $y_i$ denotes its corresponding label. Consider $T_\mathbf{e}$ is the poisoning function and $(\mathbf{x}_i, \mathbf{c}_i, y_i)$ is a clean data from the training dataset, then $T_\mathbf{e}$ is defined as:

$$T_\mathbf{e} : (\mathbf{x}_i, \mathbf{c}_i, y_i) \rightarrow (\mathbf{x}_i, \mathbf{c}_i \oplus \tilde{\mathbf{c}}, y_{tc}). \tag{33}$$

The objective of the attack is to guarantee that the compromised model $f(g(\mathbf{x}))$ functions normally when processing instances characterized by clean concept vectors, while consistently predicting the target class $y_{tc}$ when presented with concept vectors that contain the trigger $\tilde{\mathbf{c}}$. The corresponding

objective function can be summarized as follows:

$$\max_{\mathcal{D}^j \in \mathcal{D}} \Sigma_{\mathcal{D}^j}(f(\mathbf{c}_j) - f(\mathbf{c}_j \oplus \tilde{\mathbf{c}}))$$

$$\text{s.t.} \quad f(\mathbf{c}_j) = f(\mathbf{c}_j \oplus \tilde{\mathbf{c}})) = y_{tc}, \tag{34}$$

where $\mathcal{D}^j$ represents each data point in the dataset $\mathcal{D}$, $y_{tc}$ is the target class, and $\mathbf{c}_j \oplus \tilde{\mathbf{c}}$ represents the perturbed concept vector.

**Backdoor Injection.** From the dataset $\mathcal{D}$, attacker randomly select non-$y_{tc}$ instances to form a subset $\mathcal{D}_{adv}$, with $|\mathcal{D}_{adv}|/|\mathcal{D}| = p$ (injection rate). Applying $T_e : (\mathbf{x}_i, \mathbf{c}_i, y_i) \rightarrow (\mathbf{x}_i, \mathbf{c}_i \oplus \tilde{\mathbf{c}}, y_{tc})$ to each point in $\mathcal{D}_{adv}$ creates the poisoned subset $\tilde{\mathcal{D}}_{adv}$. We then retrain the CBMs with the modified training dataset $\mathcal{D}(T_\mathbf{e}) = \mathcal{D} + \tilde{\mathcal{D}}_{adv} - \mathcal{D}_{adv}$.

## Q  IMAGE2TRIGGER_C DEMO

In this section, we discuss our initial exploration of Image2Trigger_c, where we implement a preliminary demo using UNet. When applied to the CAT attack on the CUB dataset with an injection rate of 0.1 and a trigger size of 20, the ASR decrease from 59.28% to 53.29%. TSR is 72.17%, and the IS is 0.0104. These results indicate areas for improvement, particularly in terms of image similarity and ASR. Visualizations of both the original test set samples and the samples with triggers generated by Image2Trigger_c are shown in Figure 7. It is important to note that this is merely an initial demo, and we have laid the groundwork for further research by establishing a foundational pipeline. We acknowledge that there is significant room for enhancement, and we are committed to open-sourcing all the code associated with this paper upon its acceptance. We believe that this represents a novel domain, where we have not only defined a new task but also constructed a comprehensive pipeline that facilitates future research and development.

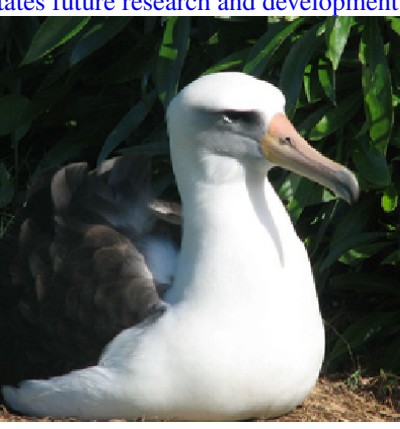 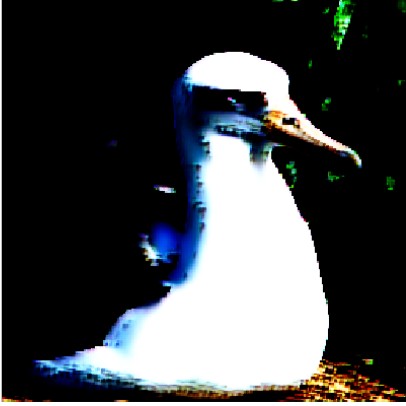

Figure 7: Visualization of the original test set samples (left) and the corresponding samples with triggers generated by Image2Trigger_c (right). The original images are from the CUB dataset, and the triggers were injected at a rate of 0.1 with a size of 20. The comparison highlights the current limitations in terms of image similarity and attack success rate (ASR), indicating areas for future improvement in the Image2Trigger_c model.

## R  EXPLAINABLE VISION ALIGNMENT OF ATTACK

We present Figure 8 for the vision alignment of our attacks here. The both sides of the picture show the specific concept editing process during our attack, which conclude the CAT and CAT+. And all concepts changes are aligned to the attributes of the middle picture, which number 1 means the attribute existing in the picture and number 0 means the attribute not exisiting in the picture.

## ETHICS STATEMENT

This work introduces and explores the concept of backdoor attacks in CBMs, a topic that inherently involves considerations of ethics and security in machine learning systems. The research aims to

| CAT | | CAT+ | |
|---|---|---|---|
| Back color is orange | 0 -> 1 | 0 -> 1 | Bill shape is curved (up or down) |
| Back color is red | 0 -> 1 | 1 -> 0 | Bill shape is spatulate |
| Under tail color is red | 0 -> 1 | 0 -> 1 | Bill shape is all-purpose |
| Nape color is orange | 0 -> 1 | 0 -> 1 | Head pattern is striped |
| Belly color is buff | 0 -> 1 | 0 -> 1 | Bill color is blue |

Figure 8: An explainable vision alignment of our attacks, the figure shows the visualizing mapping from our CAT and CAT+ attacks to the specific picture. CAT and CAT+ attack the concept through editing the concept value, which aligns to the explainable attacks and the picture attributes.

shed light on potential vulnerabilities in CBMs, with the intention of prompting further research into defensive strategies to protect against such attacks.

While our work demonstrates how CBMs can be compromised, we emphasize that the knowledge and techniques presented should be used responsibly to improve system security and not for malicious purposes. We acknowledge the potential risks associated with publishing methods for implementing backdoor attacks; however, we believe that exposing these vulnerabilities is a crucial step toward understanding and mitigating them.

Researchers and practitioners are encouraged to use the findings of this study to develop more robust and secure AI systems. It is our hope that by bringing attention to these vulnerabilities, we can collectively advance the field towards more transparent, interpretable, and secure machine learning models.

## REPRODUCIBILITY STATEMENT

To ensure the reproducibility of our results, we provide detailed descriptions of the datasets used, preprocessing steps, model architectures, and experimental settings within the paper and its appendices. Specifically:

- **Datasets:** We utilize the publicly available CUB and AwA datasets, with specific preprocessing steps outlined in Appendix H.

- **Model Architecture:** Details about the Concept Bottleneck Model (CBM) architecture, including the use of a pretrained ResNet50 and modifications for each dataset, are provided in Section I.

- **Experimental Settings:** The experimental setup, including training hyperparameters, batch sizes, learning rates, and data augmentation techniques, are thoroughly described in Section I.

- **Attack Implementation:** The methodology for implementing our proposed CAT and CAT+ attacks, including concept selection, trigger embedding, and iterative poisoning strategies, is elaborated in Sections 4 and 5, with additional insights provided in the appendices.

Furthermore, to foster transparency and facilitate further research in this area, we commit to making our code publicly available upon publication of this paper. This includes scripts for preprocessing data, training models, executing backdoor attacks, and evaluating model performance and attack effectiveness.

