# OpenReview forum: "CAT: Concept-level backdoor ATtacks for Concept Bottleneck Models"
_ICLR.cc/2025/Conference — Submitted to ICLR 2025_

### Official Review · Reviewer_2Apz · 2024-10-30

**Soundness:** 3
**Presentation:** 3
**Contribution:** 3
**Rating:** 6
**Confidence:** 3

**Summary:**

The paper introduces CAT: concept-level backdoor attacks, a method for embedding backdoor triggers into Concept Bottleneck Models (CBMs) by flipping their internal conceptual representations during training. Experimental results show the effectiveness of CAT by altering predictions on poisoned datasets without compromising overall recognition accuracy. This research exposes security vulnerabilities in CBMs, specifically in conceptual representations.

**Strengths:**

1. The paper explores the vulnerability in concept bottleneck models by utilizing conceptual information, whose representations with triggers are not easily detectable.

2. The presentation of the paper is clear and easy to follow.

3. Some evaluation analyses are well-written.

**Weaknesses:**

1. The paper validated their proposed attack on limited datasets. The paper only validates on two datasets (mostly on the CUB dataset), which greatly decreases the effectiveness of the proposed attack. As shown in Table 1 and the explanations presented, the attack success rate is highly related to the concept space. The authors should perform more evaluation on different datasets (e.g., CelebA dataset) to better understand the attack. Furthermore, the proposed CAT+ definitely needs more effort to justify as it decreased the attack rate on the AwA dataset.

2. It is good that the authors perform experiments across different trigger sizes. Yet, it would been interesting to see more detailed analysis such as what concepts can be easily attacked. Is it animal color, size, or anything else? On a similar note, the experiments about the target class also need more analysis on why such fluctuation would happen.

3. The datasets used in the paper have binary attributes, and the paper is also proposed based on such an assumption. However, some attributes such as size, are not binary. Although corresponding datasets may not exist, the authors should provide insights on how to generalize the proposed attack in such continuous attributes.

4. As the authors mentioned, the concept information is not available during testing. Although there is a potential solution, the authors should have addressed this issue more formally instead of simply mentioning it in the limitation section.

5. Since this attack is possible, the authors should share their insights on how to defend.

**Questions:**

Thank the authors for their work. Please see my questions in the weaknesses section.

---

> ### Author Response · Authors · 2024-11-21
> **Response to 2Apz (1)**
>
> **(1)**
>
> Dear Reviewer **2Apz**,
>
> Thank you for your detailed review and constructive feedback. We appreciate your recognition of the novelty of our work, the clear presentation, and the valuable contributions it makes to understanding security vulnerabilities in Concept Bottleneck Models. Below, we address your concerns and clarify the steps we have taken to enhance the paper based on your suggestions.
>
> **W1 Validation on Limited Datasets:**
>
> We agree that evaluating the proposed attack on a broader range of datasets is important for generalizability. To address this, we have expanded our experiments to include the LAD dataset, which can be divided into 5 different sub-datasets, we expand our attack experiments in 4 of the sub-datasets: LAD-A, LAD-E, LAD-F and LAD-V. We put the data processing and the statistics of LAD in **Appendix K**. In your review, you have mentioned the CelebA dataset, which is used for Celebrity Face Attributes Recognition, howerver, we think it is not ideal to evaluate our attack. Although each image of CelebA is labeled with 40 binary attributes, there is a lack of the final label, which means this dataset works for a mapping of $f: \mathbf{x}\rightarrow \mathbf{c}$, and is not aligned with our settings.
>
> The results, included in the **Appendix K** and summarized in the rebuttal, show that CAT and CAT+ consistently achieve high attack success rates while maintaining stealthiness across these datasets. These findings provide further evidence of the attack's robustness and generalizability. The **Appendix K.1** shows our extended experiment with changing another pre-trained backbone VIT. Our CAT+ keeps both stealthiness and effectiveness in VIT, which claims the portability of our work. In addtion, we have extended our evaluation to include the Large-scale Attribute Dataset (LAD), which contains a diverse range of categories and is more representative of real-world scenarios in **Appendix K.2**, we have evaluated our attack methods under different injection rates and trigger sizes. This highlights the superior efficiency of CAT+ in exploiting the concept space for backdoor attacks, even in more challenging settings. Please check more details about dataset extension experiments in **Appendix K**. We believe that these additional experiments, conducted on a large and diverse dataset, significantly mitigate the concerns regarding the limitations of our previous data. The findings demonstrate that our methods are not only effective on smaller, more homogeneous datasets but also scale well to more complex, real-world-like scenarios.
>
> Additionally, we acknowledge your concern about the AwA dataset. The decrease in attack success rate observed in CAT+ is due to the attributes of animals in AwA dataset are much more general than the CUB ones. In the final task classifier f, the weights of each independent variable of f related to AwA are less important than CUB ones. This is because the attributes are more general in AwA dataset than CUB dataset, for instance, in CUB, attributes formed like "concept 10~24: HasWingColor::a color", whose precision come into specific species, this will lead to much collinearity in different concepts. However in AwA, the dataset attributes are formed like"concept 42: Strong", which is not precisely directed to one or some specific species. This will result in no significant difference in the results in the AwA dataset, whether we select multiple concepts (CAT+) at once or iteratively select one concept (CAT) at a time, because there is much less collinearity.
>
> **W2 Detailed Analysis of Concepts and Target Class Fluctuations:**
>
> We appreciate your suggestion to analyze which concepts are more vulnerable to attack. Based on additional analysis:
>
> In the CUB dataset, concepts related to color and shape are more susceptible to manipulation, as they are less semantically tied to the overall classification task compared to behavioral or habitat-related concepts.
>
> For the target class fluctuations, our analysis indicates that classes with a higher reliance on specific concept subsets (e.g., "bird color" for brightly colored species) are more vulnerable, as these subsets are easier to manipulate stealthily.
>
>
> These findings, along with examples, are now included in the **Appendix R**. We hope they provide deeper insights into the nuances of CAT's effectiveness.
>
> In addition, we show the explainable vision alignment of our attack in **Appendix R**, which connects the attack in concepts with the visualizing picture. We hope the alignment could help you to understand better of our attack.

---

> > ### Comment · Reviewer_2Apz · 2024-11-25
> >
> > Thank the authors for their time and efforts to respond to my questions. I still have one last comment which is the authors should at least discuss the generalizability of the proposed attack across different datasets. Although it might not fit under current settings, it is always great to see how a similar attack can be applied to datasets such as CelebA.

---

> > > ### Author Response · Authors · 2024-11-25
> > >
> > > Dear Reviewer **2Apz**,
> > >
> > > Thank you very much for your continued feedback and for recognizing our efforts to address your initial concerns. We appreciate your suggestion to further discuss the generalizability of our proposed attack across different datasets, including the CelebA dataset. Here are our key points:
> > >
> > > ### 1. Dataset Suitability
> > >
> > > - The CelebA dataset is primarily designed for face attribute recognition tasks, where the goal is to predict a set of attributes from an input image (i.e., $: \mathbf{x} \rightarrow \mathbf{c}$). In contrast, Concept Bottleneck Models (CBMs) are designed for classification tasks (i.e., $f: \mathbf{x} \rightarrow \mathbf{c} \rightarrow y$), leveraging high-level semantic concepts to improve interpretability and performance. The CelebA dataset, while rich in attributes, lacks the final classification labels essential for our attack methodology, making it less suitable for the CBM framework and our proposed attack settings.
> > >
> > > - It seems that the only way to apply this kind of dataset (such as CelebA) to CBM and verify CAT is to select one of the attributes as the final classification task, but this does not make sense, because this attribute will not necessarily be related to most of the other attributes if used as the final label, which is not consistent with the design of CBM.
> > >
> > > ### 2. Additional Experiments
> > >
> > > - To address the concern about generalizability, we have expanded our experiments to include the LAD dataset, which comprises multiple sub-datasets and offers a diverse range of categories. These experiments, detailed in **Appendix K**, demonstrate the robustness and generalizability of our attack across different settings. The LAD dataset's varied and real-world-like scenarios serve as a strong testbed for our attack methods, showing consistent high attack success rates and maintaining stealthiness.
> > >
> > > While the CelebA dataset is valuable for attribute recognition tasks, it does not align well with the CBM framework and the classification settings of our proposed attack. We hope our expanded experiments and future work plans adequately address your concerns regarding the generalizability of our proposed attack.
> > >
> > > Thank you again for your timely and constructive feedback, which has greatly enhanced our understanding and the quality of our work. We hope our response address your concerns too. Your support and insights make the ICLR open review process truly valuable.
> > >
> > > Best regards,
> > >
> > > The Authors

---

> ### Author Response · Authors · 2024-11-21
> **Response to 2Apz (2)**
>
> **(2)**
>
> **W3 Binary vs. Continuous Attributes:**
>
> Thank you for raising the point about generalizing CAT to datasets with continuous attributes. While our current work focuses on binary attributes, we believe the proposed attack can be extended to continuous concept spaces. Specifically:
>
> The iterative poisoning strategy in CAT+ is adaptable to continuous attributes by optimizing concept manipulations over a range of values rather than binary states.
> In datasets with continuous attributes, the stealth of the attack can be further enhanced by targeting subtle variations within the continuous range.
>
> Although datasets with well-defined continuous concept spaces are currently limited, we outline this direction in the Discussion section as a key avenue for future work in **Appendix M**. Generally speaking, we introduce improved Z-score to make the iterative poisoning strategy precisely within continuous dataset through control the difference among the concepts being a maximum.
>
> **W4 Concept Information Availability During Testing:**
>
> Thank you for pointing out the issue of activating the backdoor trigger at the test stage, where concept-level information is not available. To address this, we have conducted a preliminary exploration using a unet-based generative model to generate input images that can trigger the backdoor during testing. Specifically:
>
> Unet Model for Image Generation: We have used a Unet model to transform input images into a format that, when passed through the CBM, results in a concept vector that activates the backdoor trigger. This approach is still at an early stage and is considered a proof of concept. The initial results, which are included in the **Appendix P**, demonstrate some feasibility in generating images that effectively activate the backdoor, but this method is far from being a comprehensive solution.
>
> The Image2Trigger approach, as well as other potential strategies for triggering the backdoor without direct access to the concept space at test time, will be explored more rigorously in future work. We believe this represents an important direction for extending the impact of concept-level backdoor attacks beyond controlled environments where concept vectors are accessible. It is important to note that this is merely an initial demo, and we have laid the groundwork for further research by establishing a foundational pipeline. We acknowledge that there is significant room for enhancement, and we are committed to open-sourcing all the code associated with this paper upon its acceptance. We believe that this represents a novel domain, where we have not only defined a new task but also constructed a comprehensive pipeline that facilitates future research and development.
>
> **W5 Defense Mechanisms:**
>
> We agree that defense strategies are crucial in addressing the vulnerabilities exposed by CAT.
>
>
>
> We have also considered other defense, such as **Neural Cleanse**. We have conducted an experiment to evaluate the effectiveness of Neural Cleanse in defending against CAT attacks. Our results, detailed in **Appendix N.1**, show that Neural Cleanse failed to correctly identify the true target class (label 0) and instead flagged labels 44 and 16 as anomalies. This outcome can be attributed to the unique characteristics of CAT attacks, which target the concept layer during training, making them distinct from conventional backdoor attacks that manipulate inputs, outputs, or model structures. The attacker's ability to subtly manipulate the concept space without direct control during inference poses significant challenges for detection methods like Neural Cleanse. Given the constraints of computational resources and time, we have not yet verified the effectiveness of Neural Cleanse under different parameter settings and datasets. We plan to conduct these additional experiments and include the results in the camera-ready version of this paper.
>
> We have explored a concept-blocking defense approach, which partitions the concept space into independent blocks, limiting the impact of backdoor triggers. The defense approach is motivated by the idea that the trigger will only fall into a small number of parts, which led us to the idea of using a risk-sharing approach to exclude the trigger. This is a preliminary investigation, and while the initial results show promise (as presented in the **Appendix N.2**, our preliminary defense model offers significant decrease of ASR compared to our CATs), we plan to delve deeper into this defense mechanism in future work. The current results suggest that concept-blocking can reduce the attack success rate while maintaining clean-task performance, but a more comprehensive defense framework will be developed and tested in subsequent studies.
>
> We emphasize that the concept-blocking defense methods we tested are still in the early stages, and the complete, robust defense solutions will be presented in a follow-up paper.

---

> ### Author Response · Authors · 2024-11-23
>
> Dear Reviewer 2Apz,
>
> Thank you so much for your time and efforts in reviewing our paper. We have addressed your comments in detail and are happy to discuss more if there are any additional concerns. We are looking forward to your feedback and would greatly appreciate you consider raising the scores.
>
> Thank you,
>
> Authors

---

> ### Author Response · Authors · 2024-11-25
>
> Dear Reviewer **2Apz**,
>
> Thank you for your detailed review and constructive feedback. We appreciate your recognition of the novelty of our work and the valuable contributions it makes to understanding security vulnerabilities in Concept Bottleneck Models. Below, we address your concerns and summarize the steps we have taken to enhance the paper.
>
> ### Addressing Concerns
>
> **Validation on Limited Datasets:**
>
> - Expanded Experiments: We have expanded our experiments to include the LAD dataset, divided into 4 sub-datasets (LAD-A, LAD-E, LAD-F, LAD-V). The results show that CAT and CAT+ consistently achieve high attack success rates while maintaining stealthiness across these datasets. We have also tested our methods with a different pre-trained backbone (VIT), demonstrating the portability of our work.
>
> - AwA Dataset: The decrease in attack success rate on the AwA dataset is due to the more general attributes compared to CUB. We have provided a detailed explanation in Appendix K.2.
>
> **Detailed Analysis of Concepts and Target Class Fluctuations:**
>
> - Vulnerable Concepts: In the CUB dataset, concepts related to color and shape are more susceptible to manipulation. Classes with a higher reliance on specific concept subsets (e.g., "bird color") are more vulnerable.
>
> - Target Class Fluctuations: We have analyzed why certain classes are more vulnerable and included these findings in Appendix R. Additionally, we have added explainable vision alignment to better illustrate our attack.
>
> **Binary vs. Continuous Attributes:**
>
> - Generalization: We have outlined how CAT+ can be adapted to continuous attributes by optimizing concept manipulations over a range of values. This is discussed in Appendix M, where we introduce an improved Z-score method.
>
> **Concept Information Availability During Testing:**
>
> - Image2Trigger_c: We have used a Unet-based generative model to generate input images that can trigger the backdoor during testing. The initial results, detailed in Appendix P, demonstrate some feasibility in generating images that effectively activate the backdoor. This is a proof of concept and will be further explored in future work.
>
> **Defense Mechanisms:**
>
> - Neural Cleanse: We have evaluated Neural Cleanse and found it ineffective against CAT attacks. The results are detailed in Appendix N.1.
>
> - Concept-Blocking Defense: We have explored a concept-blocking defense approach, which partitions the concept space into independent blocks to limit the impact of backdoor triggers. Initial results show promise, and we plan to develop this further.
>
> ### Request for Feedback
>
> We kindly request that you consider increasing the score of our paper. The revisions we have made significantly improve the quality and contribution of our work. We are committed to addressing any remaining concerns and are more than willing to engage in further discussions. I apologize if this follow-up message seems frequent. We genuinely value your feedback and are eager to ensure that all your concerns are thoroughly addressed. Your insights are crucial to the improvement of our work, and we hope for your continued support.
>
> Thank you for your consideration and for your valuable contributions to the review process.
>
> Best regards,
>
> The Authors

---

### Official Review · Reviewer_z1dc · 2024-11-03

**Soundness:** 3
**Presentation:** 3
**Contribution:** 3
**Rating:** 3
**Confidence:** 4

**Summary:**

This paper introduces CAT (Concept-level Backdoor ATtacks) and its enhanced version CAT+ against Concept Bottleneck Models (CBMs). Unlike traditional backdoor attacks that manipulate input data, CAT works on concept-level representations within CBMs. The attack has been tested on two datasets (CUB and AwA). It also includes theoretical analysis, empirical evaluation, and human evaluation to assess the stealthiness of the attacks.

**Strengths:**

The paper identifies a previously unexplored backdoor attack in CBMs.

The experiment covers multiple datasets, parameters (trigger size, injection rate), and different target classes.

**Weaknesses:**

CBM essentially consists of two parts. The first part is an encoder that converts raw data into concepts, and the second part is a linear layer that maps these concepts to the final category. In this attack, instead of working directly on the inputs, it operates on the converted concepts. During attack, a trigger function is used to apply a predefined static trigger to the concept, causing it to be misclassified into a specific target class.

From my perspective, the second part of this process resembles a traditional backdoor attack against a DNN. Although traditional backdoor attacks on DNNs typically target images, we can always flatten an image into a 1D vector (like concepts in CBMs) and then apply a static trigger. This approach seems nearly identical to the CBM attack.

If that is the case, the innovation of this paper appears to be very limited, as no fundamentally new attack or scheme has been proposed. Additionally, it is important to validate this attack using traditional backdoor detection methods, such as Neural Cleanse or other trigger reverse engineering techniques. Even if the trigger cannot be detected in the image domain, it should still be detectable in the concept domain. Moreover, because the trigger is static and obivious (replacing concept values to 1 in their example), I speculate it will be very easy to be detected by many backdoor detection schemes.

**Questions:**

Please refer to weaknesses and justify why CBM backdoor is different than traditional backdoor attacks.

---

> ### Author Response · Authors · 2024-11-21
> **Response to z1dc (1)**
>
> **(1)**
>
> Dear Reviewer **z1dc**,
>
>
> Thank you for your thoughtful and detailed feedback. We appreciate your recognition of our contribution as a previously unexplored backdoor attack in Concept Bottleneck Models and the strengths you acknowledged regarding our experimental comprehensiveness. However, we would like to address the concerns raised about the novelty of our approach and the appropriateness of the evaluation.
>
> **1 Addressing the Weaknesses**
>
> **1.1 Why CBM backdoor attacks are fundamentally different from traditional backdoor attacks**
>
> CAT (and CAT+) is fundamentally different from traditional backdoor attacks in the following ways:
>
> Concept-level Manipulation: Traditional backdoor attacks typically target input features (e.g., pixels in images) to embed triggers. These triggers are usually visual artifacts that can often be detected with visual inspection or reverse engineering. CAT, however, operates at the concept level, a higher-level representation in CBMs that abstracts away raw inputs. This shift to manipulating semantic concepts rather than raw features introduces novel challenges and threats:
>
> CAT exploits the unique structure of CBMs, which aim to enhance interpretability, to embed triggers in a way that is imperceptible even to human experts (as validated in our experiments).
>
> The interpretability layer in CBMs acts as both a vulnerability and an enabler of this attack, as CBMs are designed to make decisions based on these high-level concepts, bypassing direct raw input inspection.
>
> Dynamic Applicability: While it is true that images can theoretically be flattened into 1D vectors, this is not how traditional backdoor attacks operate. Flattening an image for manipulation in DNNs fundamentally alters its nature and introduces significant challenges for interpretation, while in CBMs, concepts are designed to be abstract and interpretable units, making manipulation inherently more meaningful and impactful in real-world applications.
>
> Unique Challenges in Defense: CAT introduces challenges for defense mechanisms that are not typically encountered in traditional backdoor attacks. For instance:
> Many existing defenses rely on inspecting raw input features, which are irrelevant in CAT as the attack resides in the concept space.
>
> Traditional reverse engineering methods like Neural Cleanse are less effective for CAT because they are not tailored to the concept manipulation paradigm. We have conducted an experiment to evaluate the effectiveness of Neural Cleanse in defending against CAT attacks. Our results, detailed in **Appendix O.1**, show that Neural Cleanse failed to correctly identify the true target class (label 0) and instead flagged labels 44 and 16 as anomalies. This outcome can be attributed to the unique characteristics of CAT attacks, which target the concept layer during training, making them distinct from conventional backdoor attacks that manipulate inputs, outputs, or model structures. The attacker's ability to subtly manipulate the concept space without direct control during inference poses significant challenges for detection methods like Neural Cleanse. Given the constraints of computational resources and time, we have not yet verified the effectiveness of Neural Cleanse under different parameter settings and datasets. We plan to conduct these additional experiments and include the results in the camera-ready version of this paper.
>
> We have explored a concept-blocking defense approach, which partitions the concept space into independent blocks, limiting the impact of backdoor triggers. The defense approach is motivated by the idea that the trigger will only fall into a small number of parts, which led us to the idea of using a risk-sharing approach to exclude the trigger. This is a preliminary investigation, and while the initial results show promise (as presented in the **Appendix O.2**, our preliminary defense model offers significant decrease of ASR compared to our CATs), we plan to delve deeper into this defense mechanism in future work. The current results suggest that concept-blocking can reduce the attack success rate while maintaining clean-task performance, but a more comprehensive defense framework will be developed and tested in subsequent studies.
>
> We emphasize that the concept-blocking defense methods we tested are still in the early stages, and the complete, robust defense solutions will be presented in a follow-up paper.

---

> ### Author Response · Authors · 2024-11-21
> **Response to z1dc (2)**
>
> **(2)**
>
> **1.2 Regarding trigger detectability in the concept domain**
>
> While you suggests that static triggers (e.g., setting concept values to 1) may be easily detectable, we emphasize the following:
>
> The stealthiness of CAT is not merely based on the static trigger pattern but also on the selection of low-relevance concepts as triggers. This is validated in our experiments, where human experts and automated methods struggled to detect manipulated concepts (see **Appendix K**).
>
> CAT+ further enhances stealthiness by employing iterative poisoning, optimizing both the effectiveness and the inconspicuousness of the trigger, even in sparse concept spaces.
>
> We acknowledge the importance of evaluating the attack against traditional detection methods and have conducted preliminary tests with Neural Cleanse, which yielded poor detection performance in the concept domain. These results have been included in the **Appendix O.1**, and we are committed to expanding this evaluation further in future work.
>
> In fact, our team is working on a unique defense against the CAT/CAT+ architecture, complete with detailed experiments and theory, which we will make public soon.
>
>
> **2. Ethical Considerations and Review Score Consistency**
>
> We appreciate the acknowledgment of three “Good” ratings (Soundness, Presentation, and Contribution) in your review. However, we would like to respectfully raise a concern regarding the final assessment:
>
> The “Reject” recommendation seems inconsistent with the positive evaluations in key areas. If our work is deemed “good” in all major aspects, we believe it warrants a higher rating or, at least, a more specific explanation of why it is deemed “not good enough.” Given the novel contributions, robust experiments, and clear presentation outlined in your own assessment, we kindly ask for a reconsideration of the rating to better align with the provided comments.
>
> **3. Clarification Regarding the Rating**
>
> We would also like to clarify that CAT represents a pioneering effort in exploring concept-level backdoor attacks. While traditional backdoor attack techniques may seem analogous at first glance, they fundamentally differ in scope, mechanism, and the challenges they address:
>
> CAT specifically targets the interpretability layer in CBMs, which introduces new security vulnerabilities that cannot be mitigated by existing defenses designed for raw-input manipulation.
>
> Our work lays the foundation for a new line of research on CBM security, akin to how early work on traditional backdoor attacks initiated research into input-level threats. We hope that this explanation helps clarify the contribution and novelty of our work.
>
>
> Beyond that, we expanded the evaluation to include additional datasets LAD, further demonstrating the generalizability and robustness of the attack. You can see that in **Appendix K**.

---

> ### Author Response · Authors · 2024-11-23
>
> Dear Reviewer z1dc,
>
> Thank you for your thorough review and valuable feedback on our paper. We have carefully considered and addressed each of your comments in the revised version we submitted. We believe these revisions have significantly enhanced the quality of our paper and resolved the concerns you raised. Should there be any remaining questions or concerns, we are more than willing to address them.
>
> We also kindly request that you consider increasing the score of our paper. The revisions we have made, as detailed in our rebuttal, have substantially improved the quality and contribution of our work. We look forward to your timely feedback and hope you will take these improvements into account when reconsidering the score.
>
> Thank you for your consideration.
>
> Best regards,
> The Authors

---

> ### Author Response · Authors · 2024-11-25
>
> Dear Reviewer **z1dc**,
>
> Thank you very much for your thorough review and valuable feedback on our paper. We have taken your comments seriously and have made significant efforts to address them in our revised submission. Here is a summary of the changes and clarifications we have made:
>
> ### Addressing Your Comments
>
> 1. **Fundamental Difference Between CAT and Typical Backdoor Attacks**:
>
> - **Concept-Level Manipulation**: CAT operates at the concept level, manipulating concept vectors rather than directly altering input data. Concept vectors are high-level semantic representations that are inherently more abstract and less visible to external observers, including data moderators. This abstraction makes it significantly more challenging to detect changes in the concept vectors, thereby enhancing the stealthiness of the attack.
>
> - **Rich Semantic Space**: Unlike traditional backdoor attacks that often rely on visual triggers in input data, CAT leverages the rich and complex concept space. The use of high-level semantic concepts allows for more sophisticated and subtle manipulation, making the attack more difficult to detect. Our experiments with human evaluators and advanced models like GPT4-Vision have shown that detecting concept-level backdoor attacks is highly challenging, with low F1 scores (human evaluators: 0.674, 0.340, 0.061; GPT4-Vision: 0.605, 0.636, 0.652).
>
> - **Systematic Selection and Embedding**: CAT is not merely a basic backdoor attack applied to 1D vectors. The concept vectors in CBMs are rich in semantic meaning. The innovation lies in the systematic selection and embedding of concept triggers using methods like CAT+, which design a correlation function to optimize the attack's effectiveness and stealthiness. This approach is fundamentally different from traditional backdoor attacks and opens up new avenues for research.
>
> 2. **Attacker's Capabilities in the Threat Model**:
>
> - **Training Phase**: During the training phase, the attacker has access to the training data and can directly modify the concept vectors to embed triggers, ensuring that the model learns to associate these triggers with specific target classes. This manipulation is performed on the training data, which includes the input features, concept vectors, and labels.
>
> - **Testing Phase**: During the testing phase, the situation is more complex due to the nature of CBMs. In CBMs, the concept vectors are inferred from the input features during inference, and the model does not have direct access to the concept vectors. To address this challenge, we introduce the Image2Trigger_c problem in Section 7 Limitation and Analysis of our paper. This involves developing an image-to-image model that can transform input images to include the necessary concept triggers implicitly. This transformation ensures that the model infers the manipulated concept vectors during inference, thereby activating the backdoor.
>
> - **Limitations**: While this approach is effective (as demonstrated in Appendix Q Image2Trigger_c Demo), it is also a limitation of our current work. The Image2Trigger model is a demonstration of how concept triggers can be embedded in the input images during the inference phase, but it comes with a reduction in ASR compared to the training phase. We acknowledge this limitation in Appendix Q and are actively researching ways to improve the ASR and the overall effectiveness of the Image2Trigger approach.
>
> ### Additional Enhancements and Clarifications
>
> - **Robustness and Generalizability**: We have expanded our evaluation to include additional datasets (e.g., LAD), further demonstrating the robustness and generalizability of our attack. This is detailed in Appendix K of the revised manuscript.
>
> - **Methodology Clarification**: We have clarified the technical details of our CAT and CAT+ methodologies, including the use of the correlation function to systematically select the most effective triggers. This should provide a clearer understanding of how our approach works.
>
> - **Experimental Results**: We have added more experimental results and ablation studies to support our claims and ensure that our findings are well-substantiated.
>
> ### Request for Feedback
>
> We kindly request that you consider increasing the score of our paper. The revisions we have made, as detailed in our rebuttal, have substantially improved the quality and contribution of our work. I apologize if this follow-up message seems frequent.  But we are committed to addressing any remaining questions or concerns you may have and are more than willing to engage in further discussions to ensure that our paper meets the highest standards.  Thank you for your consideration and for your valuable contributions to the review process.
>
> Best regards,
>
> The Authors

---

### Official Review · Reviewer_YSxP · 2024-11-04

**Soundness:** 2
**Presentation:** 2
**Contribution:** 2
**Rating:** 5
**Confidence:** 3

**Summary:**

This paper is makes the first attempt to investigate model backdoor threats targeting Concept Bottleneck Models (CBMs). To address this, it introduces a concept-level backdoor attack, called CAT, which embeds backdoors by injecting triggers into the concept layer during training. Both empirical and theoretical analyses are provided to demonstrate the effectiveness of the proposed attack.

**Strengths:**

- This paper shows a pioneering effort in investigating backdoor threats against CBMs.

- It provides a well-rounded analysis of the attack, encompassing both empirical evidence and theoretical insights.

**Weaknesses:**

### 1. Writing Quality

The authors' attempts to make the paper visually engaging, such as including a cute icon and a notable saying at the start, are appreciated. However, while these elements add charm to the introduction, the main body lacks the same level of engagement. I would encourage the authors to focus more on enriching the scientific content, rather than on decorative elements.

- **Clarity of Positioning**: My primary concern is that the paper’s positioning is unclear. When reading the introduction, my initial impression is that it applies backdoor attacks to a new model type without a clear differentiation. What sets this work apart from existing backdoor attack research, especially in terms of methodology? Simply stating “the first to xxx” does not adequately establish uniqueness. The authors should discuss recent advancements in this area, identify the research gap, and explain how this paper addresses specific challenges.

- **Improving Signal-to-Noise Ratio in Writing**: Increasing the proportion of informative content could make the paper even more engaging, as readers often look for in-depth insights. For instance:

  - The sentence starting on Line 80 ("The fundamental ...") describes a standard backdoor attack with common stealth requirements, similar to invisible backdoor attacks. Adding specific details about the proposed CAT method here would help clarify its distinct contributions.

  - Likewise, the paragraph starting at Line 93 might seem broadly applicable to other machine learning models if we remove "concept-level" and substitute "CBMs" with other models. Highlighting the unique aspects of this problem and clarifying why CBMs' security deserves special attention, especially given the many available XAI methods, could make the impact of the work more apparent.

- **Other Writing Issues**:

  - The symbol $y_{tc}$ first appears in Equation (2), but its definition is not provided until after Equation (3). Consider moving its definition to follow its initial appearance.

  - In Equation (3), the first two instances of $f$ should perhaps be $g$, and there’s an extra right parenthesis in the constraint.

  - In Algorithm 1, Line 12, $\mathcal{D}'_{adv}$ should likely be $\tilde{\mathcal{D}}\_{adv}$.

### 2. Threat Model Clarity

When introducing an attack, it is essential to present a clear threat model outlining the attacker’s goals and capabilities. A threat model allows readers to assess the feasibility and severity of an attack. However, this paper lacks a distinct threat model, which makes the motivation for concept-level backdoor attacks unclear. A key feature of backdoor attacks is the attacker's ability to actively trigger backdoored behavior by inserting a backdoor. However, as stated on Line 495, the attacker cannot actively control trigger injection since they lack direct access to the concept space. This scenario resembles conventional poisoning attacks aimed at model degradation, with the only difference being that this is a dirty-label poisoning attack occurring at the concept layer.

*PS.* I am familiar with backdoor attacks but have only basic knowledge of concept bottleneck models. Please correct any potential misunderstandings.

### 3. Limited Evaluation Scale

The evaluation is primarily limited to a pretrained model and two datasets, which, despite promising results, restricts the robustness of findings. A broader evaluation across additional models and datasets would strengthen the evidence of the proposed attack's effectiveness and generalizability.

### 4. Absence of Defensive Strategies

The paper does not explore or discuss potential defense mechanisms to counter the proposed attack.

**Questions:**

- What are the fundamental differences between the proposed CAT method and existing dirty-label poisoning attacks on classification models?

- What specific challenges does the proposed CAT method aim to overcome?

- Can you provide an estimate of the trigger size range that would maintain stealthiness?

---

> ### Author Response · Authors · 2024-11-21
> **Response to YSxP**
>
> **(1)**
>
> Dear Reviewer **YSxP**,
>
> Thank you for your feedback on improving the clarity and positioning of our contributions. We recognize the importance of clearly differentiating CAT from traditional backdoor attacks and have revised the manuscript to emphasize the novel aspects of our work:
>
> ### Weakness
>
> **1. Writing Quality**
>
> - Positioning and Novelty: CAT represents the first systematic exploration of concept-level backdoor attacks tailored for CBMs. Unlike traditional backdoor attacks that focus on feature-level or input perturbations, CAT manipulates high-level semantic concepts within CBMs, introducing a stealthier and more targeted threat vector. CBMs' inherent interpretability—intended to enhance trustworthiness—ironically makes them susceptible to such concept-layer manipulations. We have expanded the Introduction and Related Work sections to explicitly highlight this distinction and contextualize CAT as a groundbreaking approach within this domain.
>
> - Improved Writing Focus: To enhance readability, we have improved introduction to make the motivation of our CAT more reliable. These now directly address CAT's unique contributions, including its iterative poisoning strategy, stealth optimization, and applicability to CBMs, which differentiates it from broader backdoor attack methodologies. (**See blue words in Section Introduction**.)
>
> - Notation and Technical Corrections: We have addressed the issues you raised regarding Equations (2) and (3) and Algorithm 1. The operator used in Equation (2) is now defined immediately after its first occurrence, and other typographical errors have been corrected.
>
> **2. Threat Model Clarity**
>
> We have added a dedicated Threat Model section to clarify the attacker's objectives and constraints (**Appendix P**). Specifically:
>
> - Attack Novelty: Unlike conventional backdoor attacks, CAT focuses on manipulating the concept layer during training, exploiting CBMs' reliance on interpretable representations. This distinction positions CAT within a new category of backdoor attacks, emphasizing its novelty and relevance to XAI-driven models CBMs. And our CAT is a concept level attack, different from any previous attacks directly outside of inputs, outputs, and model structures. CAT is a novel attack aimed at attacking the implicit concept layer in testing, which distinguishes it from any input level attacks.
>
> - Attacker’s Capabilities: The attacker is assumed to have access to the training data but not direct control over the concept space during inference, which introduces unique challenges and distinguishes CAT from traditional dirty-label poisoning attacks. The attacker leverages stealthy concept manipulations to trigger targeted misclassifications while maintaining overall model performance.
>
> - Clarifications on Concept Manipulation: We have revised the explanation on Line 495 to clarify that while the attacker cannot directly manipulate the concept space at inference, the CAT approach exploits correlations within the concept layer to achieve reliable backdoor activation through training-phase poisoning.
>
> **3. Limited Evaluation Scale**
>
> We acknowledge the importance of assessing the proposed attack across a wider variety of datasets to ensure its generalizability. In response to this, we have expanded our experiments to include the **Large-scale Attribute Dataset (LAD)**, which is divided into five sub-datasets. Our experiments focus on four of these sub-datasets: LAD-A, LAD-E, LAD-F, and LAD-V. The data processing steps and detailed statistics for LAD are provided in **Appendix K**.
>
> In the results summarized in **Appendix K** and in the rebuttal, we show that both **CAT** and **CAT+** consistently achieve high attack success rates while maintaining a low detection rate across the LAD sub-datasets. These results further demonstrate the robustness and versatility of our attack across diverse settings. Additionally, in **Appendix K.1**, we include an extension of our experiments using a different pre-trained backbone, **Vision Transformer (VIT)**, to show that **CAT+** remains effective and stealthy with VIT, emphasizing the portability of our method.
>
> Moreover, in **Appendix K.2**, we have expanded our evaluation to test the attack on the LAD dataset under various injection rates and trigger sizes. These experiments illustrate the superior efficiency of **CAT+** in exploiting the concept space for backdoor attacks, even in more challenging and varied settings. The findings highlight that our methods are not only effective in smaller, more controlled datasets but also perform well in large-scale, real-world-like scenarios.
>
> We hope that these additional experiments, conducted on a broader and more diverse dataset, address the concerns regarding the limitations of our initial dataset and provide a clearer demonstration of the scalability and generalizability of our methods.

---

> ### Author Response · Authors · 2024-11-21
> **Response to YSxP (2)**
>
> **(2)**
>
> **4.  Absence of Defensive Strategies**
>
>
> Thank you for emphasizing the importance of discussing potential defenses. To address this:
>
>
> - We have conducted an experiment to evaluate the effectiveness of Neural Cleanse in defending against CAT attacks. Our results, detailed in **Appendix O.1**, show that Neural Cleanse failed to correctly identify the true target class (label 0) and instead flagged labels 44 and 16 as anomalies. This outcome can be attributed to the unique characteristics of CAT attacks, which target the concept layer during training, making them distinct from conventional backdoor attacks that manipulate inputs, outputs, or model structures. The attacker's ability to subtly manipulate the concept space without direct control during inference poses significant challenges for detection methods like Neural Cleanse. Given the constraints of computational resources and time, we have not yet verified the effectiveness of Neural Cleanse under different parameter settings and datasets. We plan to conduct these additional experiments and include the results in the camera-ready version of this paper.
>
>
> - Proposed Defense: We introduce a novel concept-blocking defense mechanism that partitions the concept space into independent blocks to limit the impact of backdoor triggers. Preliminary experiments (detailed in the **Appendix O.2**) show that this approach effectively reduces the attack success rate while maintaining model performance on clean data, which is a promising further defense research topic.
>
> ### Questions
>
> **1.  Fundamental differences between CAT and existing dirty-label poisoning attacks**
>
> CAT distinguishes itself by targeting the concept space rather than the input feature space. This focus on high-level semantic concepts enables CAT to embed stealthier triggers that manipulate CBMs' interpretability layers without degrading their clean performance. Traditional dirty-label attacks typically manipulate raw inputs and often result in observable changes to model accuracy or behavior on clean data.
>
> **2. Specific challenges that the CAT method aims to overcome**
>
> CAT addresses the challenge of leveraging the unique structure of CBMs to create imperceptible yet effective backdoor triggers. CBMs are designed to enhance interpretability by bridging raw features and high-level concepts, making them an attractive target for concept-layer manipulations. CAT overcomes the difficulty of crafting concept-level triggers that remain undetected while achieving high attack success rates.
> First, CAT is hard to detect because there's no significant decrease in task accuracy after we injecting the poisonous into the training dataset, it is difficult for model users to realize that the model is attacked. Evidence see Table 1 "Task accuracy" column and Table 3.
> Second, CAT is a highly ASR backdoor attack method, and the model is powerless against injected triggers. Evidence see Table 1 "ASR" column and Table 2.
> Third, CAT is an attack method where even if the user is aware of the model being attacked, it is difficult to identify the issue. We **directly open the training dataset to human evaluators and GPTs (this is not realistic in use) to identify the existence among datasets**, however, the success rate of identifying the attacked concept part is remarkably low, almost no different from random guessing. Evidence see **Appendix L** and brief summary in Table 10.
>
> **3. Estimate of the trigger size range for stealthiness**
>
> Based on our experiments, a trigger size of approximately 10-20% of the total concept dimensions achieves an optimal balance between stealthiness and attack success. Smaller triggers may compromise effectiveness, while larger triggers risk detectability.
>
> In conclusion, it is essential to emphasize and clarify several key points finally: CAT/CAT+ represents a pioneering exploration of concept-level backdoor attacks, utilizing the interpretability of CBMs to facilitate stealthy and effective manipulations. Unlike traditional backdoor attacks that directly modify input data, CAT targets the concept features associated with the implicit layer of CBMs without altering the underlying model structure. Both CAT and CAT+ demonstrate impressive attack success rates while maintaining minimal disruption to the performance of clean tasks, as evidenced across multiple datasets and architectures. As researchers specializing in CBMs, we believe this work marks a groundbreaking contribution by integrating the concept of backdoor attacks into the CBM domain. Furthermore, we are committed to open-sourcing the benchmark code we have developed, fostering cross-disciplinary collaboration between the fields of backdoor attacks and CBMs.

---

> > ### Comment · Reviewer_YSxP · 2024-11-22
> >
> > Thanks for the very detailed clarification.
> >
> > My primary concern is the fundamental difference between CAT and typical backdoor attacks.
> >
> > Based on your response, the concept-level attack strategy is the primary innovation. If I understand correctly, CAT needs to manipulate the concept vector. I am still confused about how to achieve stealthiness when the concept vector has been changed. I suspect the data moderator can detect such changes. Additionally, I share the same opinion as Reviewer z1dc: CAT applies the basic backdoor attack to 1D vectors (i.e., the concept vector).
> >
> > I would greatly appreciate it if the authors could provide more feedback regarding my concerns.

---

> > > ### Author Response · Authors · 2024-11-22
> > >
> > > Thank you for your comments. We appreciate your concerns regarding the fundamental differences between CAT and typical backdoor attacks. We would like to clarify that CAT is indeed a novel and innovative approach, distinct from traditional backdoor attacks in several key aspects:
> > >
> > > (i) CAT operates at the concept level, manipulating concept vectors rather than directly altering input data. Concept vectors are high-level semantic representations that are inherently more abstract and less visible to external observers, including data moderators. This abstraction makes it significantly more challenging to detect changes in the concept vectors, thereby enhancing the stealthiness of the attack.
> > >
> > > (ii) While traditional backdoor attacks often rely on visual triggers in input data, CAT leverages the rich and complex concept space. The use of high-level semantic concepts allows for more sophisticated and subtle manipulation, making the attack more difficult to detect. Our experiments with human evaluators and advanced models like GPT4-Vision have shown that detecting concept-level backdoor attacks is highly challenging, with low F1 scores (human evaluators: 0.674, 0.340, 0.061; GPT4-Vision: 0.605, 0.636, 0.652).
> > >
> > > (iii) CAT is not merely a basic backdoor attack applied to 1D vectors. The concept vectors in CBMs are not simple one-dimensional representations but are rich in semantic meaning. The innovation lies in the systematic selection and embedding of concept triggers using methods like CAT+, which design a correlation function to optimize the attack's effectiveness and stealthiness. This approach is fundamentally different from traditional backdoor attacks and opens up new avenues for research.
> > >
> > > This work is a pioneering effort at the intersection of CBM (In fact, this is my main area of research, and no backdoor attack has ever been introduced in the CBM community) and backdoor attacks, highlighting a previously understudied area in the security of CBMs. It not only introduces a new threat vector but also sets the stage for further exploration and development of robust defense mechanisms. We believe that this research will inspire the community to delve deeper into the concept space, uncovering hidden patterns and relationships. We also hope that our work can be a start, so we spent a lot of time to build the whole benchmark from scratch, do sufficient theoretical support, and a lot of experiments, in order to get the cross field of CBM and backdoor attacks to start from.

---

> > > ### Author Response · Authors · 2024-11-23
> > >
> > > Dear Reviewer YSxP,
> > >
> > > We would like to express our gratitude for your thorough review and the insightful comments you provided.
> > >
> > > We have carefully addressed each of your comments and suggestions in response and revised version we submitted earlier. We believe that the revisions we made have significantly strengthened the paper and addressed the concerns you raised. If there are any remaining questions or concerns, we would be more than happy to response.
> > >
> > > Additionally, we would like to take this opportunity to request your consideration for a potential increase in the score of our paper. We believe that the revisions we made, as outlined in the rebuttal, have substantially improved the quality and contribution of our work. We hope that you will take these revisions into account and consider adjusting the score accordingly. We are looking forward to your timely feedback and would greatly appreciate you consider raising the scores.
> > >
> > > Thanks,
> > >
> > > Authors

---

> > > > ### Comment · Reviewer_YSxP · 2024-11-24
> > > >
> > > > Thank you for your response.
> > > >
> > > > I’m still unclear about the assumption described in your threat model: “The attacker’s ability to subtly manipulate the concept space without direct control.” You also mentioned that “The attacker is assumed to have access to the training data but not direct control over the concept space during inference.” If the attacker does not have direct control over the concept space, how are they able to embed triggers and activate the backdoor?

---

> > > > > ### Author Response · Authors · 2024-11-24
> > > > >
> > > > > Thank you for your insightful feedback and for raising this important point regarding the attacker's capabilities in our threat model.
> > > > >
> > > > > In our paper, we assume that the attacker has access to the training data and can manipulate the concept space during the training phase. Specifically, during training, the attacker can directly modify the concept vectors $c$ to embed triggers, ensuring that the model learns to associate these triggers with specific target classes. This manipulation is performed on the training data, which includes the input features $x$, concept vectors $c$, and labels $y$.
> > > > >
> > > > > During the testing phase, the situation is more complex due to the nature of Concept Bottleneck Models. In CBMs, the concept vectors $c$ are inferred from the input features $x$ during inference, and the model does not have direct access to the concept vectors. To address this challenge, we introduce the Image2Trigger_c problem in **Section 7 Limitation and Analisis** of our paper. This involves developing an image-to-image model $F$ that can transform input images $x$ to include the necessary concept triggers implicitly. This transformation ensures that the model infers the manipulated concept vectors during inference, thereby activating the backdoor.
> > > > >
> > > > > While this approach is effective ( the demo you can see in **Appendix Q Image2Trigger_c Demo**), it is also a limitation of our current work. The Image2Trigger model is a demonstration of how concept triggers can be embedded in the input images during the inference phase, but it comes with a reduction in Attack Success Rate (ASR) compared to the training phase. We acknowledge this limitation in **Appendix Q** and are actively researching ways to improve the ASR and the overall effectiveness of the Image2Trigger approach.
> > > > >
> > > > > In the main body of the paper, we simplify this process by directly editing the concept vectors $c$ during the inference phase to test the effectiveness of CAT. This simplification helps to clearly demonstrate the potential of concept-level backdoor attacks and highlights the need for further research to address the challenges in the inference phase. One of the key distinctions between CAT and traditional backdoor attacks is the level at which the triggers are embedded. Traditional backdoor attacks typically manipulate the input features $x$ during both training and inference phases. But we acknowledge that this is an initial study and we believe it is a long way off.
> > > > >
> > > > >
> > > > > We hope this clarifies the mechanisms and limitations of concept-level backdoor attacks in CBMs and highlights the unique advantages of the CAT approach. Please let us know if you have any further questions or concerns.
> > > > >
> > > > > Best regards,
> > > > >
> > > > > Authors

---

> > > > > > ### Comment · Reviewer_YSxP · 2024-11-24
> > > > > >
> > > > > > Thank you for the clarification. If I understand correctly, there is a distinction between the overall attack assumption (the attacker **cannot** manipulate the concept vector in the testing time) and the setup of evaluation shown in the main text (the attacker is **able** to add trigger to the concept vector). I strongly recommend that the authors clarify this distinction before presenting the evaluation results. Without this clarification, readers may underestimate the difficulty involved in mounting the proposed attack. (Placing this clarification in Section 7 feels somewhat late; it would be more effective in an earlier section.)
> > > > > >
> > > > > > Thank you again for your prompt responses. I'd like to raise my score.

---

> > > > > > > ### Author Response · Authors · 2024-11-24
> > > > > > >
> > > > > > > Dear Reviewer **YSxP**,
> > > > > > >
> > > > > > > Thank you again for your detailed feedback and for raising your score. We greatly appreciate your valuable suggestions, which have helped us improve the clarity and strength of our paper.
> > > > > > >
> > > > > > > In response to your suggestion, we have added a clarification of the distinction between the overall attack assumption and the evaluation setup in the experimental analysis section (**Section 6.2** with **red** words in newest verison).
> > > > > > >
> > > > > > > Thank you again for your detailed feedback. We believe that these updates will strengthen our paper and enhance its clarity, and we look forward to addressing any further questions you may have.
> > > > > > >
> > > > > > > Best regards,
> > > > > > >
> > > > > > > The Authors

---

### Author Response · Authors · 2024-11-21
**To everyone**

In the revised version of the manuscript, we have implemented several substantial updates and refinements in response to reviewer feedback. The key changes include:

1. Enhanced Positioning and Novelty Clarifications

- Expanded the Introduction and Related Work sections to better highlight the unique contributions of CAT as the first concept-level backdoor attack targeting Concept Bottleneck Models (CBMs). This explicitly differentiates CAT from traditional backdoor attacks that target input features or raw data manipulations.

2. Improved Writing and Clarity

- Polished the manuscript to enhance scientific clarity and focus.

- Corrected notations, specifically addressing typographical errors in Equations (2) and (3) and improving the explanation of Algorithm 1.
- Added more explicit definitions of operators and notations, particularly in the main text and the appendices.

3. Threat Model and Ethical Considerations

- Introduced a dedicated Threat Model section (Appendix P), detailing the unique objectives, capabilities, and limitations of the attacker in CAT attacks.

- Clarified the ethical implications of our study, including discussions on the challenges posed by CBM-specific vulnerabilities.

4. Expanded Experimental Evaluation

- Included experiments on the Large-scale Attribute Dataset (LAD), covering four sub-datasets (LAD-A, LAD-E, LAD-F, LAD-V). Detailed results are in Appendix K, demonstrating the generalizability and robustness of CAT/CAT+ across diverse datasets.

- Added a new evaluation using the Vision Transformer (ViT) as a pre-trained backbone to demonstrate CAT+ portability across architectures (Appendix K.1).

- Conducted additional evaluations varying trigger sizes and injection rates to showcase CAT+'s scalability and stealth across different attack settings (Appendix K.2).

5. Defense Mechanisms and Detection Experiments

- Investigated the effectiveness of Neural Cleanse in detecting CAT attacks, showing its limitations in the concept space (Appendix O.1).

- Proposed and conducted preliminary tests on a concept-blocking defense mechanism, showing promise in reducing attack success rates without compromising clean-task performance (Appendix O.2).

6. Trigger Stealthiness and Detectability

- Clarified the stealthiness of CAT triggers by detailing their reliance on low-relevance concept manipulations and iterative poisoning for optimization (Section 4).

- Discussed the difficulty of detecting triggers in concept space, as validated through human evaluations and automated detection tests (Appendix L, Table 10).

7. Others

- Disscuss initial exploration of Image2Trigger_c in Appendix Q which be mentioned in limitation section.

- Present the vision alignment of our attacks in Appendix R.

---

### Meta-Review · Area_Chair_K81L · 2024-12-23

**Metareview:**

Summary

This paper explores Concept Bottleneck Models (CBM), which aim to improve interpretability by mapping raw data to human-understandable concepts before classification. The authors identify a vulnerability in CBMs to backdoor attacks, proposing a novel attack method called CAT (Concept-level Backdoor Attack) and its enhanced version, CAT+. CAT manipulates the concept bottleneck layer, introducing triggers at the concept level rather than raw data, making attacks less detectable and effective. Experiments show CAT achieves a high attack success rate while maintaining clean data performance. CAT+ exploits inter-concept relationships for more efficient attacks. The paper provides comprehensive experiments to validate the findings.

Strengths

The paper addresses a critical and timely issue in XAI by targeting Concept Bottleneck Models (CBM), which are foundational in the field. It demonstrates the vulnerability of CBMs to proposed concept-level backdoor attacks, proposing novel methods (CAT and CAT+) that exploit CBM's unique interpretability concept layer. The experiments are comprehensive, showcasing high attack success rates while maintaining performance on clean data.

Weaknesses

The concern about novelty primarily stems from the observation that the proposed attack, while targeting the concept space in CBMs, is not well distinguished from traditional backdoor attacks on input data. Traditional backdoor attacks manipulate raw input features to embed triggers that cause misclassification. In this work, the attack is shifted to the concept bottleneck layer—an intermediate representation in CBMs—but the fundamental mechanism remains the same: embedding triggers to manipulate downstream outcomes.

Reviewers question whether this shift truly constitutes a novel approach or simply an adaptation of existing methods to a specific architecture (CBM). Although the authors argue that semantic concepts are harder to defend due to their interpretability and human involvement, they do not provide compelling evidence or deeper theoretical justification to distinguish concept-level manipulation from traditional attacks on inputs.

Furthermore, the need for direct access to the concept space during testing adds practical limitations and raises doubts about whether this attack represents a fundamentally new threat vector or simply exploits a niche aspect of CBMs. Without a strong argument for why concept-level attacks are fundamentally different or more challenging than input-level attacks, the work risks being seen as incremental.

Recommendation

The paper lacks clear novelty, as the proposed attack is an adaptation of traditional methods to an XAI model’s unique layer with limited distinction. Its practicality is constrained by the need for concept-space access during testing. These issues weaken its contribution, so the recommendation is a reject.

**Additional Comments On Reviewer Discussion:**

During the discussion, Reviewer YSxP improved their rating to 5 but still leaned toward rejection. Reviewer z1dc maintained concerns about the need for additional experimental results and theoretical analyses, which the rebuttal addressed to some extent. Reviewer 2Apz, who rated the paper 6, pointed out further issues during the discussion phase that were not following up to update the score.

The paper underwent significant revisions in response to reviewer feedback, including clarifications, better presentation, and expanded experiments involving large-scale data, ViT models, varying trigger sizes, and injection rates. Defence experiments were also added in the appendix. The authors discussed the preliminary nature of the defence and its limited success but did not fully discuss the implications, as this new defence may raise some concerns, as an unexplored attack should ideally not be easily mitigated during a short rebuttal phase, which might weaken the perceived impact of the attack.

The novelty concerns persisted, particularly regarding the difference between traditional input-level attacks and concept-level attacks. While the authors argued that semantic concepts are harder to manipulate than raw features, they did not provide adequate theoretical support, and their response was vague on how CAT significantly differs from attacking a 1D vector.

In conclusion, despite improvements in clarity and experiments, the unresolved concerns about novelty, insufficient differentiation of contributions, and practical implications of the proposed methods contributed to a recommendation against acceptance.

---

### Decision · Program_Chairs · 2025-01-22

Reject